# IN-CONTEXT LEARNING WITHOUT COPYING

## ABSTRACT

Induction heads are attention heads that perform inductive copying by matching patterns from earlier context and copying their continuations verbatim. As models develop induction heads, they often experience a sharp drop in training loss, a phenomenon cited as evidence that induction heads may serve as a prerequisite for more complex in-context learning (ICL) capabilities. In this work, we ask whether transformers can still acquire ICL capabilities when inductive copying is suppressed. We propose HAPAX, a setting where we omit the loss contribution of any token that can be correctly predicted by induction heads. Despite a significant reduction in inductive copying, performance on abstractive ICL tasks (i.e., where the answer is not contained in the input context) remains comparable and achieves higher accuracy than the vanilla model on 12 out of 18 statistically significant tasks, even though 31.7% of tokens are omitted from the loss. Furthermore, our model achieves lower loss values on token positions that cannot be predicted correctly by induction heads. Mechanistic analysis further shows that models trained with HAPAX develop fewer and weaker induction heads but still preserve ICL capabilities. Taken together, our findings indicate that inductive copying is not essential for learning abstractive ICL mechanisms.

## 1 INTRODUCTION

Language modeling is fundamentally repetitive: in a coherent document, many sequences appear more than once, like "the Dursleys" in Harry Potter, or "public static void" in Java code. Words that *do* appear only once in a given text are so rare that they are given a special name by corpus linguists: *hapax legomena*, a transliteration of Ancient Greek for "said once". But what happens if a Large Language Model (LLM) is trained such that every $n$-gram within its context window is a previously unseen *hapax legomena*?

In previous work, Elhage et al. (2021) showed that LLMs develop *induction heads* that perform inductive copying by matching patterns and copying them from earlier context, with Olsson et al. (2022) hypothesizing that these circuits provide a foundation for more complex in-context learning (ICL) capabilities. However, subsequent work has demonstrated that induction heads actually operate in parallel with different components (Feucht et al., 2025; Todd et al., 2024; Hendel et al., 2023) that are more causally important for performance on high-complexity ICL tasks (Yin & Steinhardt, 2025).

This motivates our central question: if we suppress a model's ability to copy earlier sequences from context (inductive copying), can it still learn ICL capabilities? We introduce the HAPAX training regime in which repeated $n$-grams ($n > 1$) within a context window do not contribute to the loss. Under this constraint, the model never receives gradient signals from repeated $n$-grams, meaning that it can never use induction heads to predict the tokens it is being trained on. Consequently, HAPAX models' performance on a verbatim repetition task drops by 66% relative to the vanilla model. Interestingly, we find that for the statistically significantly different tasks, HAPAX models achieve higher accuracy in 12 out of 18 and 20 out of 21 tasks across our two evaluation settings. Our findings suggest that inductive copying is causally not essential for models to develop abstractive ICL capabilities.

Figure 1: Demonstration of the induction circuit. Previous token heads allow each token to store which token came previously. Induction heads do a match-and-copy operation to reproduce the subsequence that appeared earlier in the context.

## 2 BACKGROUND

Despite being trained to do simple next-token prediction, LLMs exhibit the impressive capability to do in-context learning (ICL) (Brown et al., 2020; Chowdhery et al., 2023), where they are able to perform tasks demonstrated in-context "on-the-fly" without additional training. Olsson et al. (2022) present the first mechanistic analysis of ICL capabilities in LLMs and identify *induction circuits* as a fundamental mechanism. Induction circuits consist of three steps: (1) *previous token heads* that allow each token to store which token came before it, (2a) *induction heads* that attend to the previous token information in earlier contexts, resulting in a "prefix-matching" attention pattern, and (2b) increasing the probability of the attended token in the output. Step (2b), where the head increases the probability of the attended token, is what we refer to as inductive copying: the reproduction of token sequences that appeared earlier in the context (Figure 1). Formally, given input tokens $(x_1, \ldots, x_j)$, induction circuits operate by searching for tokens that hold information of the current token $x_j$ (i.e., searching $x_{i+1}$ where $x_i = x_j$, $i < j$) (Elhage et al., 2021). If there is a matching $x_{i+1}$, the induction head increases the logit of $x_{i+1}$ for the next prediction.

Olsson et al. (2022) observe that some of these heads are also involved in "fuzzy" copying based on semantic similarity. Feucht et al. (2025) show that models contain separate concept induction circuits that are causally more important than traditional induction circuits for "fuzzy" copying tasks (e.g., translation). In a similar vein, Yin & Steinhardt (2025) show that ablation of traditional induction heads does not damage ICL as much as ablation of *function vector* heads (Todd et al., 2024), suggesting that the latter may be more important for ICL.

Both synthetic and natural language setups have shown that the development of induction circuits is associated with rapid phase transitions as it starts to acquire prefix-matching and inductive copying capabilities (Edelman et al., 2024; Reddy, 2024; Olsson et al., 2022). After the model rapidly learns induction heads, it also experiences a general increase in ICL capabilities later in training for natural language setups. Theoretical work has identified data distributional properties that drive the emergence of in-context learning Chan et al. (2022) and shown the importance of repetition throughout data distribution for emergent behavior of large language models Zucchet et al. (2025). Building on prior work, we examine the importance of induction heads' ability to perform inductive copying for in context learning. Leveraging these insights, we directly manipulate the training data distribution in a natural language setting such that there is no incentive to learn inductive copying, and observe the resultant model's ICL capabilities.

## 3 HAPAX

### 3.1 TRAINING PROTOCOL

To suppress inductive copying, we apply loss masking to create a training regime where tokens that can be correctly predicted by induction heads are excluded from the loss calculation. Previous work has used loss masking to exclude the loss contributions of certain tokens to prevent memorization of private information (Hans et al., 2024; Kosireddy & Lucas, 2025). Here, we use the same approach to remove the incentive for models to learn inductive copying. Specifically, we mask the loss contributions of token positions that contain a matching $n$-gram within the same context window (where $n > 1$). Single-token repetitions are not masked because they cannot be predicted by induction. Thus, the first token of any repeated $n$-gram is left unmasked. Importantly, masked tokens are still visible to all model components and they are only excluded from the loss computation.

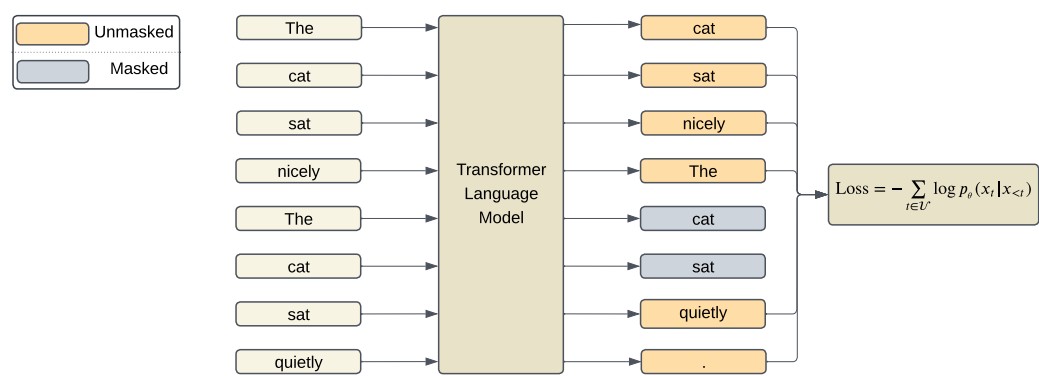

Figure 2: An overview of HAPAX training regime. To suppress inductive copying, we introduce HAPAX where positions predictable by induction heads within a context window do not contribute to the loss (gray positions). This discourages n-gram repetitions in the training distribution, allowing us to control the strength of inductive copying and observe its effect on in context learning.

Formally, let $\mathbf{x} = (x_t)_{t \in S}$ be an input sequence of tokens from vocabulary $\mathcal{V}$, where $S = \{1, 2, \ldots, T\}$ denotes the set of all token positions and $x_t \in \mathcal{V}$ for all $t \in S$. The probability assigned by the model to the correct prediction at a position $t$ is defined as:

$$p_\theta(x_t | x_{<t}) = \text{softmax}(f_\theta(x_{<t}))_{x_t} \qquad (1)$$

where $f_\theta$ denotes the model with parameters $\theta$. We define the set of masked positions as:

$$\mathcal{M} = \{j \in S : \exists i < j \text{ such that } (x_{i-1}, x_i) = (x_{j-1}, x_j)\} \qquad (2)$$

The set of unmasked positions is then $\mathcal{U} = S \setminus \mathcal{M}$. We compute the masked loss, a negative log-likelihood, only over the unmasked positions:

$$\mathcal{L}_{\text{masked}} = -\frac{1}{|\mathcal{U}|} \sum_{t \in \mathcal{U}} \log p_\theta(x_t | x_{<t}) \qquad (3)$$

This means that the HAPAX model is never trained on any tokens that can be correctly predicted using induction heads, removing the incentive to learn inductive copying.

We train vanilla and HAPAX 1B models from scratch using the transformers library (Wolf et al., 2020) implementation of GPT-NeoX (Andonian et al., 2023). We use the same hyperparameter and training configuration as the Pythia models (Biderman et al., 2023). We use the Pile dataset to train our models (Gao et al., 2020). We train our models for 20000 steps, which we observed empirically to be sufficient to analyze the emergence of in-context learning dynamics. We save model checkpoints every 100 training steps. The vanilla model is trained on 40B tokens whereas the HAPAX model is trained on 28B tokens due to loss masking. Further training variants that we explored but found ineffective are provided in Appendix E.

### 3.2 SIMILARITY-THRESHOLDED HAPAX

Certain token pairs can have a high cosine similarity in the input embedding space. As an example, the tokens "National" and " National" (with a leading space) have a cosine similarity of 0.84 in the input embedding space of the vanilla model. Although they are not exactly the same token, the model can still acquire a high copying signal through these similar tokens. Referring back to Equation 2, we modify the exact matching with a thresholded matching logic in order to create a stricter suppression. We mask a position if the corresponding tokens have a cosine similarity higher than $\tau$:

$$\mathcal{M} = \{j \in S : \exists i < j \text{ such that } S_{cos}(e_{x_{i-1}}, e_{x_{j-1}}) > \tau \text{ and } S_{cos}(e_{x_i}, e_{x_j}) > \tau\} \qquad (4)$$

where $e_{x_i}$ refers to the embedding for the token $x_i$. We choose $\tau = 0.3$. Consequently, the model is trained on 19B tokens. See Appendix B for details.

## 4 EFFECTS OF HAPAX ON ICL

We now test the ICL capabilities of HAPAX. In the following sections we verify that the HAPAX methodology is able to suppress inductive copying capabilities. We observe a reduction for *extractive* ICL task performance as described by Todd et al. (2024). An extractive task is a few-shot task where the model must directly extract the answer from the input, (e.g., "foil car purple : foil, pen cloud window :" → "pen"). However, when we analyze *abstractive* tasks (Todd et al., 2024), which require generating new answers rather than copying (e.g., "Greece : Athens, China : Beijing, Egypt" → "Cairo"), we observe that HAPAX preserves abstractive mechanisms and achieves higher accuracy on 12 out of 18 statistically significant tasks. These results provide evidence that models can learn abstractive ICL without relying on inductive copying, and the incentive to not repeat previous tokens might be beneficial for a subset of these tasks. In Appendix C, we complement these findings with a small evaluation on the broader effect of HAPAX training on natural text generation, showing an increase in natural language fluency.

### 4.1 SUPRESSION OF INDUCTIVE COPYING

We first measure random repetition performance: the model is given 1000 sequences of random repeated tokens $r_1 r_2 \ldots r_s r_1 r_2 \ldots r_{s-1}$ and is expected to predict $r_s$. This synthetic task does not occur in natural language but is solvable through induction heads and the same sequence is also used to identify prefix-matching attention patterns Nanda & Bloom (2022). Figure 3a demonstrates that HAPAX causes a major drop in random repetition performance. The HAPAX model experiences a 66% drop and Thresholded-HAPAX experiences an 89% drop in accuracy relative to the vanilla model at the end of training. We also evaluate performance on natural text repetition in Figure 3b, where the models are given 1000 sequences of the form $r_1 r_2 \ldots r_s r_1 r_2 \ldots r_{s-1}$ with $r_1 r_2 \ldots r_s$ taken from whole sequences in the WikiText dataset Merity et al. (2017). Natural text repetition accuracy decreases over time, suggesting the model's increased incentive to not repeat natural text. The initial increase of natural text repetition is due to the model's increased language modeling capacity, as a non-repeated sequence $r_1 r_2 \ldots r_{s-1}$ has the same initial accuracy for the HAPAX model (Figure A9).

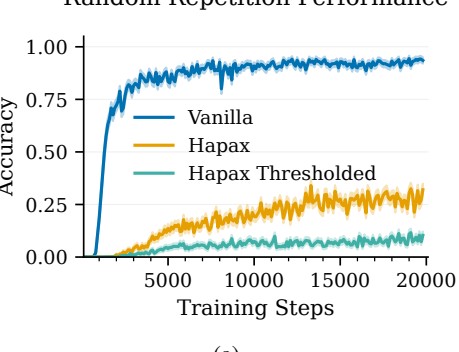 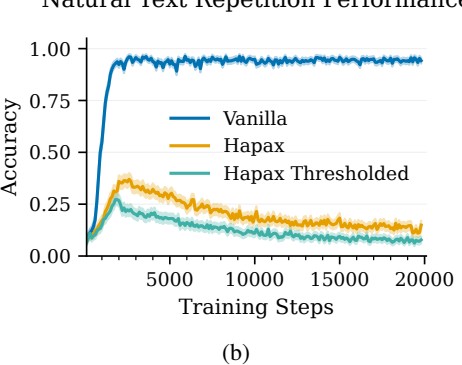

(a)                                           (b)

Figure 3: Repetition performance in both random token and natural text settings. (a) HAPAX models struggle with repeating random sequences of tokens, a task that is solvable with induction circuits. (b) For natural text repetition, HAPAX models actively suppress copying as training progresses. Accuracy is measured over 1000 randomly generated samples with $s = 25$.

We evaluate HAPAX on 28 extractive tasks from Todd et al. (2024) and find statistically significant changes on 19, with 17 showing reduced performance (Table A1). The results confirm that the HAPAX training regime effectively reduces inductive copying and repetition. We present the results for Thresholded-HAPAX in Appendix A.

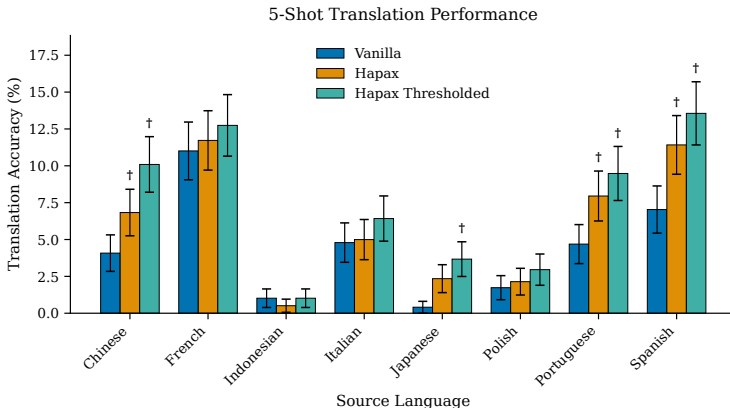

Figure 4: Word-level translation performance across all 3 models. We observe that both HAPAX and Thresholded-HAPAX increases the translation performance. This indicates that reduction in inductive copying does not hinder the model's ability to learn translation. The translations are from a given source language into English in order to keep the number of predicted tokens same across different languages. Error bars indicate 95% binomial normal-approximation CIs. † denotes statistical significant difference ($p < 0.05$) with vanilla model accuracy according to McNemar's test.

## 4.2 PERFORMANCE PRESERVATION FOR ABSTRACTIVE ICL

We now continue our evaluation on abstractive tasks, where the model needs to generate novel information not contained in the context. We evaluate on 23 abstractive tasks from Todd et al. (2024) (e.g., Capitalize First Letter) and 8 word-level translation tasks. All tasks are evaluated in a 5-shot setting. As shown in Figure 4 and Table 1, the model on average maintains its performance. Of the 23 abstractive tasks, HAPAX achieves higher accuracy on 9 of the 15 tasks with statistically significant differences, and is at least comparable to the vanilla model on 17 tasks overall. When we include the translation tasks, HAPAX achieves higher accuracy on 12 out of 18 significant tasks. We note that when using HAPAX, 31.7% of the tokens are masked and the preservation of performance is visible despite being trained on 31.7% fewer tokens. Thresholded-HAPAX performs worse for majority of abstractive tasks (Table A2) than vanilla model as the masking is much stricter with 52.5% of the tokens being masked. Interestingly, Thresholded-HAPAX achieves higher accuracy than HAPAX and the vanilla model on 7 of 8 tasks, with 4 statistically significant. We attribute the contrasting trends between the translation task and other abstractive tasks under Thresholded-HAPAX to our choice of threshold. A cosine similarity threshold of 0.3 masks many same-language tokens, while cross-language tokens typically fall below this threshold and therefore contribute more training signal. See Appendix B for a discussion on the choice of threshold.

Additionally, for the abstractive tasks from Todd et al. (2024), we observed that some tasks have small label spaces, causing the target token to frequently appear in the 5-shot context. To ensure the vanilla model is not matching the context's label distribution, we reran the abstractive evaluation using only few-shot examples that exclude the target token. Under this control, the HAPAX model achieves higher accuracy than the vanilla model on 17 out of 18 significant tasks (Table A4). THRESHOLDED-HAPAX also improves, achieving higher accuracy for 10 out of 22 significant tasks, a substantial improvement over the previous results (Table A5). These suggest that the performance of the vanilla model in abstractive tasks sometimes resulted from distributional copying of labels from the context.

Overall, if abstractive ICL mechanisms were fundamentally dependent on inductive copying, we would expect performance degradation across most tasks. However, our results do not show such degradation. The disincentive of our HAPAX model to learn exact repetition has not caused significant damage, and has even increased performance for a subset of these tasks, despite being trained on much fewer tokens.

Table 1: Performance comparison on **abstractive** in-context learning tasks (5-Shot). Values show accuracy $\pm$ 95% CI margin. Bold indicates higher performance; † denotes statistical significance ($p < 0.05$) according to McNemar's test.

| Task | Vanilla (%) | HAPAX (%) | Task | Vanilla (%) | HAPAX (%) |
|---|---|---|---|---|---|
| Capitalize First Letter | $37.3 \pm 3.3$ | $\mathbf{68.3 \pm 3.2}^{\dagger}$ | Singular-Plural | $67.8 \pm 6.4$ | $\mathbf{80.5 \pm 5.4}^{\dagger}$ |
| Country-Capital | $31.5 \pm 6.5$ | $\mathbf{43.7 \pm 6.9}^{\dagger}$ | AG News | $\mathbf{32.9 \pm 1.3}^{\dagger}$ | $8.9 \pm 0.8$ |
| Lowercase First Letter | $\mathbf{72.2 \pm 3.1}$ | $69.0 \pm 3.2$ | Synonym | $1.6 \pm 0.5$ | $\mathbf{2.6 \pm 0.6}^{\dagger}$ |
| CommonsenseQA | $\mathbf{19.8 \pm 1.1}^{\dagger}$ | $5.9 \pm 0.7$ | Capitalize Second Letter | $\mathbf{11.7 \pm 2.2}^{\dagger}$ | $2.2 \pm 1.0$ |
| Landmark-Country | $32.9 \pm 3.2$ | $\mathbf{36.5 \pm 3.3}^{\dagger}$ | Product-Company | $\mathbf{20.1 \pm 3.4}$ | $19.2 \pm 3.4$ |
| Capitalize Last Letter | $\mathbf{9.7 \pm 2.0}^{\dagger}$ | $3.7 \pm 1.3$ | Antonym | $0.8 \pm 0.3$ | $\mathbf{2.3 \pm 0.6}^{\dagger}$ |
| Lowercase Last Letter | $6.8 \pm 1.7$ | $\mathbf{7.1 \pm 1.8}$ | Word Length | $\mathbf{7.9 \pm 1.8}$ | $6.9 \pm 1.7$ |
| Next Item | $12.0 \pm 4.2$ | $\mathbf{31.1 \pm 6.0}^{\dagger}$ | National Parks | $20.4 \pm 3.7$ | $\mathbf{23.5 \pm 3.9}$ |
| Country-Currency | $29.9 \pm 6.4$ | $\mathbf{32.5 \pm 6.5}$ | Capitalize (Full Word) | $\mathbf{76.9 \pm 2.9}^{\dagger}$ | $58.2 \pm 3.4$ |
| Present-Past | $54.3 \pm 5.7$ | $\mathbf{68.6 \pm 5.3}^{\dagger}$ | Previous Item | $5.8 \pm 3.0$ | $\mathbf{8.0 \pm 3.5}$ |
| Park-Country | $12.4 \pm 2.4$ | $\mathbf{17.0 \pm 2.7}^{\dagger}$ | Next Capital Letter | $\mathbf{6.4 \pm 1.7}$ | $5.9 \pm 1.6$ |
| Sentiment | $\mathbf{65.3 \pm 2.7}^{\dagger}$ | $17.4 \pm 2.2$ | | | |

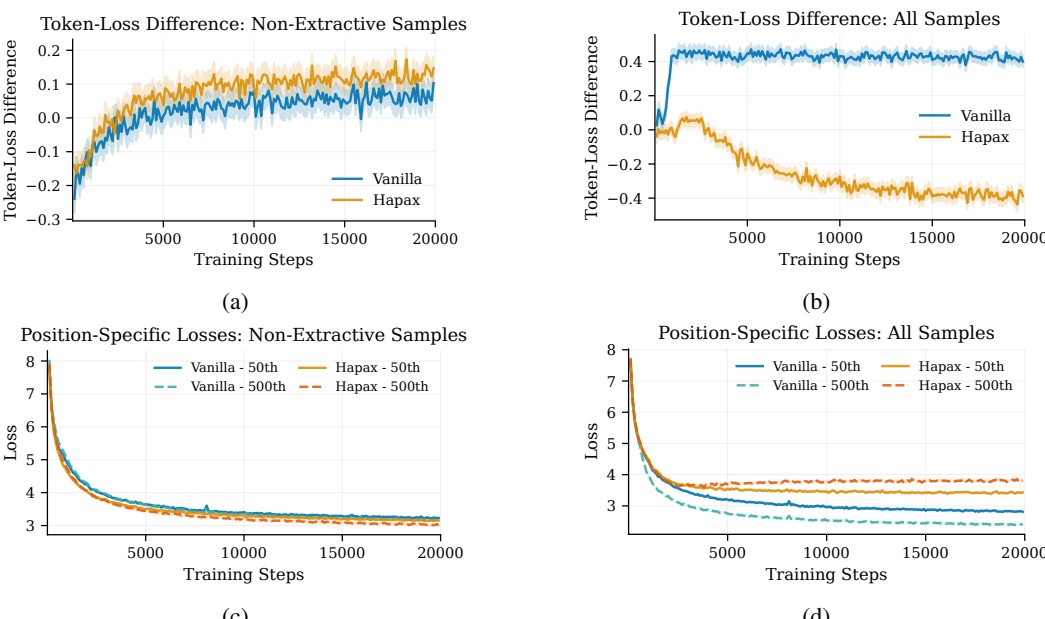

(a)      (b)

(c)      (d)

Figure 5: Comparison of token-loss metrics and positional loss values across conditions. The HAPAX model exhibits lower loss values and better TLD scores for token positions that cannot be predicted by induction. However, it receives a negative score when we consider all samples. This indicates that the loss-dependent metrics mostly capture the gains from exact copying strategy but does have implications for non-extractive samples. The metrics are calculated on randomly sampled data from the validation dataset.

### 4.3 IN-CONTEXT LEARNING BEYOND n-GRAM COPYING

Building on the results from Sections 4.1 and 4.2, which suggest that models do not depend on inductive copying to acquire more abstract ICL capabilities, we now analyze loss values of HAPAX and vanilla models throughout training. We use the token-loss difference metric (termed by Yin & Steinhardt (2025) to understand general ICL capabilities. This metric is defined by the differences of the cross-entropy loss of arbitrary token positions, conventionally using the 500th and 50th token positions. Formally, let $L_t = -\log p_\theta(x_t \mid x_{<t})$ denote the cross-entropy loss at token position $t$. The token-loss difference (TLD) between the 50th and 500th tokens is then given by $\Delta TLD = L_{50} - L_{500}$. Intuitively, token-loss difference measures improvement across increasing token positions. If the loss at token 500 is lower and $TLD > 0$, it shows that the model's predictions improved with increasing context. Yin & Steinhardt (2025) provided evidence that the metric is strongly influenced

by induction heads but does not correlate well with in-context learning task performance. Figure 5b shows that the HAPAX model experiences a significant drop in token-loss difference, getting worse over time. As HAPAX model performs worse at exact copying tasks but maintains its performance on abstract copying tasks, our results demonstrate that the sudden increase in token-loss difference metric for vanilla model is indicative of how the model learns inductive copying, but should not be taken as a measure of abstractive learning capability, which is consistent with observations from Lv et al. (2025). In natural language, n-grams are often repeated, and the token-loss difference metric is primarily affected by these exact copying instances. Because our loss masking removes 31.7% of tokens, it implies that exactly 31.7% of positions in the data are those that induction could have correctly predicted through copying. This statistic is implied by the extension of Zipf's Law to $n$-grams Ha et al. (2009). All these results provide evidence that the exact copying capability and the phase shifts in loss-dependent metrics gained by induction heads can be explained primarily by the distributional attributes of natural language, but it does not have implications for abstractive capabilities.

To investigate this hypothesis further, we propose using samples where neither the 500$^{th}$ nor the 50$^{th}$ token can be predicted correctly with inductive copying. With this modification, we will be able to understand the improvement of ICL for tokens that cannot be predicted with inductive copying. Figure 5a shows that, contrary to the regular token-loss difference metric, the HAPAX model has a slightly higher token-loss difference, which suggests that it can leverage context better for non-exact copying instances and aligns with results from Section 4.2. We also observe that the model's ability to leverage context for non-exact matching tokens does not exhibit a phase shift but rather improves gradually across training steps. Figure 5c shows that not only the difference but also the individual losses at these positions are lower in the loss masked model. This demonstrates that learning to leverage context for non-exact copying instances (as calculated by the TLD score on non-extractive samples) has improved, despite the reduction in inductive copying.

# 5 MECHANISTIC INVESTIGATION OF INDUCTION HEADS

## 5.1 INFLUENCE OF PREFIX MATCHING HEADS ON COPYING

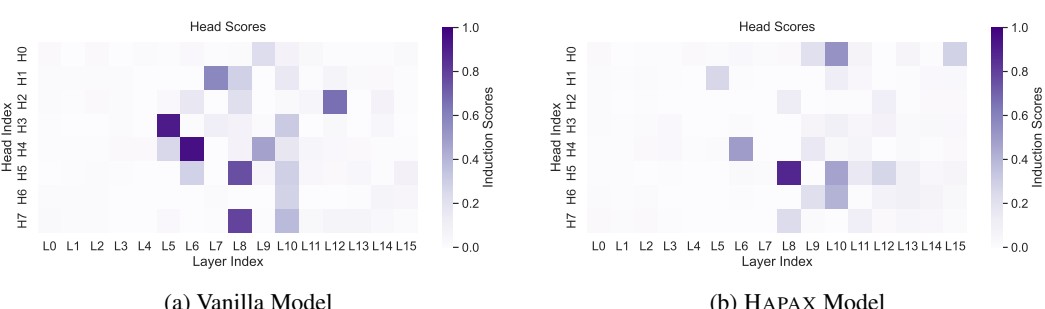

(a) Vanilla Model                                      (b) HAPAX Model

Figure 6: Prefix Matching Scores across Vanilla and HAPAX models. HAPAX model has fewer heads that have a strong prefix-matching score.

With inductive copying suppressed, we next investigate mechanistically how induction heads are affected. We analyze the attention patterns of attention heads for the vanilla and HAPAX models using the random repetition sequence discussed in Section 4.1. To obtain prefix matching score over a sequence $x = (r_1 r_2 \ldots r_s r_1 r_2 \ldots r_{s-1})$, we calculate $\text{PrefixMatching}(l, h) = \frac{1}{s-1} \sum_{i=1}^{s-1} A_{s+i,i+1}^{(l,h)}$ over 1000 samples, where $s = 25$ and $A^{l,h}$ is the attention map of attention head at layer $l$ and index $h$ Nanda & Bloom (2022). HAPAX has fewer attention heads that strongly display the prefix-matching pattern commonly associated with induction heads (Figures 6a and 6b). In the vanilla model, the top 10 prefix-matching heads achieve an average score of 61%, whereas in HAPAX this average drops to 40%. Also, there is only a single attention head (L8H5) that attends with a near 100% score. As noted in Section 4.1, other head types can also exhibit a similar pattern. We therefore analyze the prefix-matching heads to quantify their contribution on inductive copying.

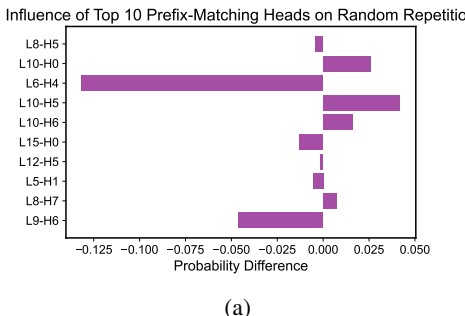

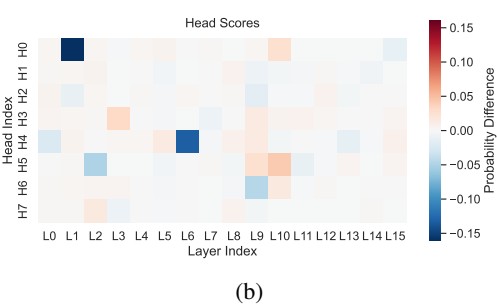

(a)             (b)

Figure 7: Influence of individual attention heads for inductive copying. Different from the vanilla model, many of the top 10 prefix-matching heads are negatively influencing copying, as shown in (a).

Using the same setup, we conduct ablation studies to understand the causal impact of each individual attention head for the expected prediction $r_s$. Mean ablation is a method that replaces the activation of a model component with its average activation from a reference distribution (Wang et al., 2023). For each head, we compute the mean activation over 10000 samples from the Pile dataset that the model was trained on (Gao et al., 2020). We only mean ablate the activations of the last token position in order to isolate the behavior of induction heads (see Appendix A.2). We look at the probability difference of the target token $r_s$ before and after ablation, computing $p_{clean}(r_s|x) - p_{ablated}(r_s|x)$

Figure 7b shows the probability differences for each head after mean ablating. To get a clearer view of how prefix matching determines copying behavior, we display in Figure 7a the probability difference scores of the top 10 heads that have the highest prefix matching scores with their rankings preserved. We observe from Figure 7a that out of the top 10 prefix matching heads, 6 of the heads negatively influence the probability assigned to correct token, meaning that they functionally behave closer to an anti-induction head (McDougall et al., 2024; Olsson et al., 2022). Despite many of the top 10 prefix-matching heads negatively influencing prediction, abstract ICL capabilities remain intact. This suggests that learning abstractive ICL relies less on inductive copying. A reduction in inductive copying and induction heads did not compromise the learning of abstractive capabilities for the HAPAX model.

## 5.2 Cross-Checkpoint Patching Experiments

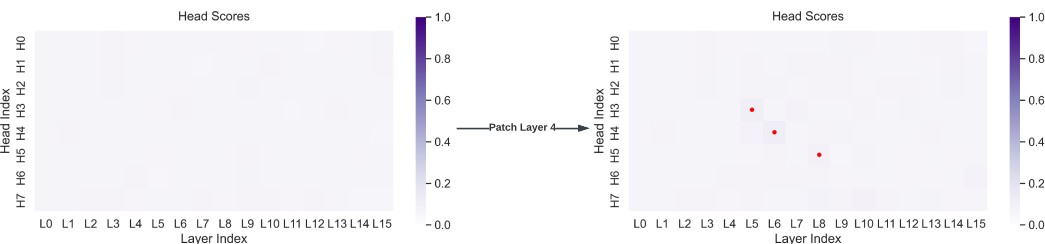

Figure 8: Induction head scores of the randomly initialized vanilla model before and after patching layer L4 of the vanilla model at step 5000, which contains the first previous-token heads (Figure A11). The maximum induction score on a randomly repeated sequence is 0.076, while patching layer L4 raises the maximum to 0.122. Heads marked with red dots are the top three prefix-matching heads which also rank among the top prefix-matching heads in the vanilla model's final checkpoint, showing that some heads are biased toward prefix matching from initialization and once previous token heads are formed, certain heads are naturally going to attend to the flowing information.

With HAPAX training, we obtained a model that does not benefit from repetition. However, our data distribution plausibly does not imply anything about the existence of previous token heads, and they might still be helpful for tasks such as detokenization (Lad et al., 2025; Gurnee et al., 2023;

Feucht et al., 2024; Kaplan et al., 2025). In this section, we conduct another set of experiments to ascertain the influence of previous token heads on the formation of induction heads. If models must develop previous token heads for reasons other than learning induction circuits, heads in later layers may naturally develop prefix-matching attention patterns as they attend to this information, which is why we might still see reduced but non-zero inductive copying behavior. Here, we find that even *randomly initialized* heads at later layers will attend to previous token information, suggesting that prefix-matching patterns can form as a direct result of the presence of previous token information.

We analyze two models: our vanilla model at step 5000, and the same model at step 0 (random initialization). At step 5000, our vanilla model has previous token heads at layer L4, but no induction heads appear in layer L4 or earlier (Figure A11). We hypothesize that when this previous token information is present, randomly initialized heads in later layers will begin to exhibit prefix-matching attention patterns. Therefore, we apply activation patching by patching the outputs of layer L4 from step 5000 into layer L4 at step 0, and observe whether heads in later layers attend to this previous token information despite being randomly initialized.

Figure 8 shows that once we patch the outputs of layer L4 at step 5000 into the randomly-initialized model, the model's highest induction score increases to 0.122. (Before patching, the randomly-initialized model had a maximum prefix matching score of 0.076.) All heads highligted with red dot for this experiment also rank among the top 5 prefix matching heads at the final checkpoint of the vanilla model (Figure 6a), indicating that these heads were biased towards prefix matching from initialization. A similar pattern holds for the HAPAX model, where 2 out of 3 top prefix-matching heads from its final checkpoint also have increased induction scores in our cross-checkpoint patching experiment (L8H5, L6H4). Both the HAPAX and vanilla model start training from the same random initialization seed, which can explain why we are seeing this overlap. These results suggest that induction head-like attention patterns can form quite easily once previous token information is present, possibly explaining how the HAPAX model is still able to learn such attention patterns despite never being trained on token positions that can be predicted by induction heads.

## 6 RELATED WORK

**Attention Heads and ICL** Early work identified induction circuits responsible for inductive copying (Elhage et al., 2021) and analyzed their influence on ICL (Olsson et al., 2022; Crosbie & Shutova, 2025), hypothesizing they are the basis of in-context learning. Subsequent studies found various attention heads important for ICL, including function vector heads that trigger tasks (Todd et al., 2024), concept induction heads that copy lexical units (Feucht et al., 2025), semantic induction heads extracting relations (Ren et al., 2024), symbolic induction heads inducing over abstract variables (Yang et al., 2025), and n-gram generalizations of induction heads (Akyürek et al., 2024). Prior work also noted correlations between task-specific heads and canonical induction heads, questioning whether the latter are necessary for learning ICL-related attention heads. We show that despite weaker induction heads and reduced copying capacity, models maintain robust abstract ICL capabilities that do not require exact copying.

**Training Dynamics and Induction Heads.** Theoretic work on synthetic setups has provided insights into induction head development and training dynamics. Several studies analyze phase transitions where Singh et al. (2024) use clamping to study subcomponents of induction heads and their effect on phase changes. Chan et al. (2022) shows how skewed Zipfian distributions lead to the emergence of ICL. Bietti et al. (2023) demonstrates the evolution from global bigram statistics to induction head solutions over training, and Chen et al. (2024); Edelman et al. (2024) further analyze convergence to induction-like solutions in Markov chain data with Edelman et al. (2024) demonstrating the different phases where the model learns more complex strategies with more training. Minegishi et al. (2025) studies a meta-learning setting to investigate mechanisms in non-exact copying scenarios. In our work, we instead focus on a natural language setting and create a data distribution where induction heads can never predict the correct next token.

**Loss Masking.** Prior work incorporated loss masking strategies to prevent memorization for information of interest (Hans et al., 2024; Kosireddy & Lucas, 2025). We incorporate a loss masking strategy to create a distribution which allows us to suppress induction behavior throughout training.

**Repetition.** Prior work has studied repetition from the perspective of data distribution and model outputs. Zucchet et al. (2025) analyzes how training data repetition speeds up emergent behavior in language models. Several works analyze the "repetition curse", with some finding induction heads to be causing such behavior (Hiraoka & Inui, 2025; Wang et al., 2025; Yao et al., 2025). Welleck et al. (2020) presents unlikelihood loss training to mitigate repetitions in the model. As repetition is closely tied to induction heads, our HAPAX training regime discourages repeated sequences, aiming to suppress inductive copying and, as a result, repetition.

## 7 CONCLUSION

In this work, we introduce the HAPAX training regime, which removes the incentive for the inductive copying strategy typically learned by induction heads early in training. We find that while HAPAX models struggle more with inductive copying and develop fewer induction heads, their abstractive ICL capabilities are preserved, and in some cases even improved. Our analysis shows that inductive copying is not essential for learning abstractive ICL mechanisms. We also observe weaker induction circuits, which suggests their role in learning abstractive ICL capabilities could be more limited than previously assumed. We observe that reducing repetition incentives does not adversely affect the abstractive behavior and has benefits for quality of the generation. More broadly, our methodology offers a systematic way to analyze model learning dynamics and can be adapted to explore how different attributes of the data distribution may shape learning dynamics.

## ETHICS STATEMENT

This paper aims to advance the foundational understanding of in-context learning mechanisms. While such research may influence future model development and deployment, we cannot meaningfully anticipate these downstream impacts within the scope of this work.

## REPRODUCIBILITY STATEMENT

We release training and evaluation code and data to enable replication of our findings. This includes environments, configurations (covering tokenizer, architecture, and optimizer settings), and specific seeds used for data shuffling and evaluation. The code also implements our HAPAX objective, including similarity-thresholded masking, and provides evaluation prompts and mechanistic analysis. For patching experiments, we used NNSIGHT (Fiotto-Kaufman et al., 2025).

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

# A    ADDITIONAL MODEL RESULTS

## A.1    ICL TASK RESULTS

## A.2    ABLATION STUDIES

While conducting mean ablations, we only ablate the last token position of the attention heads. We choose the last token to avoid disrupting the influence of the rest of the circuit, mainly the previous token heads. Given a sequence $r_1 r_2 ... r_s r_1 r_2 ... r_{s-1}$ the previous token head that is going to hold information for prefix matching is going to be stored at token position $r_s$, which is the end of the first half of the repetition. If we were to ablate all token positions, we would also disrupt the information flow from previous token heads, which would mistakenly lead us to identify such heads as induction heads. We also include the figures of vanilla and Thresholded-HAPAX for comparison with the HAPAX model. We chose to use probability difference metric as it correlated most accurately with task accuracy.

Table A1: Performance comparison on **extractive** in-context learning tasks (5-Shot) for Vanilla and HAPAX models. Values show accuracy $\pm$ 95% CI margin. Bold indicates higher performance; † denotes statistically significance ($p < 0.05$) according to McNemar's test.

| Task | Vanilla (%) | HAPAX (%) | Task | Vanilla (%) | HAPAX (%) |
|---|---|---|---|---|---|
| Choose First of 3 | $95.0 \pm 1.4^{\dagger}$ | $30.7 \pm 2.9$ | Choose First of 5 | $95.9 \pm 1.2^{\dagger}$ | $27.0 \pm 2.8$ |
| Choose Middle of 3 | $40.2 \pm 3.0^{\dagger}$ | $21.5 \pm 2.5$ | Choose Middle of 5 | $26.7 \pm 2.7^{\dagger}$ | $7.4 \pm 1.6$ |
| Choose Last of 3 | $66.1 \pm 2.9^{\dagger}$ | $53.6 \pm 3.1$ | Choose Last of 5 | $52.2 \pm 3.1$ | $58.7 \pm 3.1^{\dagger}$ |
| Alphabetically First (3) | $34.4 \pm 2.9^{\dagger}$ | $28.1 \pm 2.8$ | Alphabetically First (5) | $19.9 \pm 2.5^{\dagger}$ | $14.5 \pm 2.2$ |
| Alphabetically Last (3) | $32.3 \pm 2.9^{\dagger}$ | $24.5 \pm 2.7$ | Alphabetically Last (5) | $18.5 \pm 2.4^{\dagger}$ | $13.0 \pm 2.1$ |
| CoNLL: Person | $73.2 \pm 1.5^{\dagger}$ | $25.8 \pm 1.4$ | CoNLL: Location | $59.4 \pm 1.4^{\dagger}$ | $17.5 \pm 1.1$ |
| CoNLL: Organization | $57.8 \pm 1.6^{\dagger}$ | $21.2 \pm 1.3$ | Animal vs Object (3) | $48.4 \pm 3.1^{\dagger}$ | $27.3 \pm 2.8$ |
| Animal vs Object (5) | $36.3 \pm 3.0^{\dagger}$ | $16.3 \pm 2.3$ | Fruit vs Animal (3) | $45.1 \pm 3.1^{\dagger}$ | $21.5 \pm 2.5$ |
| Fruit vs Animal (5) | $28.1 \pm 2.8^{\dagger}$ | $9.9 \pm 1.9$ | Color vs Animal (3) | $63.3 \pm 3.0^{\dagger}$ | $18.6 \pm 2.4$ |
| Color vs Animal (5) | $47.0 \pm 3.1^{\dagger}$ | $9.5 \pm 1.8$ | Adjective vs Verb (3) | $57.3 \pm 3.1$ | $56.1 \pm 3.1$ |
| Adjective vs Verb (5) | $52.3 \pm 3.1^{\dagger}$ | $39.4 \pm 3.0$ | Verb vs Adjective (3) | $70.5 \pm 2.8^{\dagger}$ | $33.6 \pm 2.9$ |
| Verb vs Adjective (5) | $61.3 \pm 3.0^{\dagger}$ | $20.8 \pm 2.5$ | Concept vs Object (3) | $56.8 \pm 3.1^{\dagger}$ | $35.9 \pm 3.0$ |
| Concept vs Object (5) | $45.7 \pm 3.1^{\dagger}$ | $18.3 \pm 2.4$ | Object vs Concept (3) | $73.2 \pm 2.7^{\dagger}$ | $58.4 \pm 3.1$ |
| Object vs Concept (5) | $68.9 \pm 2.9^{\dagger}$ | $51.9 \pm 3.1$ | SQuAD Validation | $17.7 \pm 1.1^{\dagger}$ | $8.5 \pm 0.8$ |

Table A2: Performance comparison on **abstractive** in-context learning tasks (5-Shot) for Thresholded-HAPAX. Values show accuracy $\pm$ 95% CI margin. Bold indicates higher performance; † denotes statistical significance ($p < 0.05$) according to McNemar's test.

| Task | Vanilla (%) | HAPAX THRESH (%) | Task | Vanilla (%) | HAPAX THRESH (%) |
|---|---|---|---|---|---|
| Capitalize First Letter | $37.3 \pm 3.3^{\dagger}$ | $26.9 \pm 3.0$ | Singular-Plural | $67.8 \pm 6.4^{\dagger}$ | $39.5 \pm 6.7$ |
| Country-Capital | $31.5 \pm 6.5$ | $32.0 \pm 6.5$ | AG News | $32.9 \pm 1.3^{\dagger}$ | $1.1 \pm 0.3$ |
| Lowercase First Letter | $72.2 \pm 3.1^{\dagger}$ | $48.2 \pm 3.4$ | Synonym | $1.6 \pm 0.5$ | $4.4 \pm 0.7^{\dagger}$ |
| CommonsenseQA | $19.8 \pm 1.1^{\dagger}$ | $0.0 \pm 0.0$ | Capitalize Second Letter | $11.7 \pm 2.2^{\dagger}$ | $4.1 \pm 1.4$ |
| Landmark-Country | $32.9 \pm 3.2^{\dagger}$ | $21.8 \pm 2.8$ | Product-Company | $20.1 \pm 3.4^{\dagger}$ | $9.0 \pm 2.5$ |
| Capitalize Last Letter | $9.7 \pm 2.0^{\dagger}$ | $4.1 \pm 1.4$ | Antonym | $0.8 \pm 0.3$ | $7.8 \pm 1.1^{\dagger}$ |
| Lowercase Last Letter | $6.8 \pm 1.7^{\dagger}$ | $3.2 \pm 1.2$ | Word Length | $7.9 \pm 1.8^{\dagger}$ | $1.2 \pm 0.8$ |
| Next Item | $12.0 \pm 4.2$ | $10.2 \pm 4.0$ | National Parks | $20.4 \pm 3.7$ | $19.7 \pm 3.7$ |
| Country-Currency | $29.9 \pm 6.4^{\dagger}$ | $12.2 \pm 4.6$ | Capitalize (Full Word) | $76.9 \pm 2.9^{\dagger}$ | $46.1 \pm 3.4$ |
| Present-Past | $54.3 \pm 5.7^{\dagger}$ | $40.3 \pm 5.6$ | Previous Item | $5.8 \pm 3.0^{\dagger}$ | $1.8 \pm 1.7$ |
| Park-Country | $12.4 \pm 2.4$ | $15.2 \pm 2.6^{\dagger}$ | Next Capital Letter | $6.4 \pm 1.7$ | $4.5 \pm 1.4$ |
| Sentiment | $65.3 \pm 2.7^{\dagger}$ | $0.0 \pm 0.1$ | | | |

Table A3: Performance comparison on **extractive** in-context learning tasks (5-Shot) for Thresholded-HAPAX . Values show accuracy $\pm$ 95% CI margin. Bold indicates higher performance; † denotes statistical significance ($p < 0.05$) according to McNemar's test.

| Task | Vanilla (%) | HAPAX THRESH (%) | Task | Vanilla (%) | HAPAX THRESH (%) |
|---|---|---|---|---|---|
| Choose First of 3 | $95.0 \pm 1.4^{\dagger}$ | $30.7 \pm 2.9$ | Choose First of 5 | $95.9 \pm 1.2^{\dagger}$ | $27.0 \pm 2.8$ |
| Choose Middle of 3 | $40.2 \pm 3.0^{\dagger}$ | $21.5 \pm 2.5$ | Choose Middle of 5 | $26.7 \pm 2.7^{\dagger}$ | $7.4 \pm 1.6$ |
| Choose Last of 3 | $66.1 \pm 2.9^{\dagger}$ | $53.6 \pm 3.1$ | Choose Last of 5 | $52.2 \pm 3.1$ | $58.7 \pm 3.1^{\dagger}$ |
| Alphabetically First (3) | $34.4 \pm 2.9^{\dagger}$ | $28.1 \pm 2.8$ | Alphabetically First (5) | $19.9 \pm 2.5^{\dagger}$ | $14.5 \pm 2.2$ |
| Alphabetically Last (3) | $32.3 \pm 2.9^{\dagger}$ | $24.5 \pm 2.7$ | Alphabetically Last (5) | $18.5 \pm 2.4^{\dagger}$ | $13.0 \pm 2.1$ |
| CoNLL: Person | $73.2 \pm 1.5^{\dagger}$ | $25.8 \pm 1.4$ | CoNLL: Location | $59.4 \pm 1.4^{\dagger}$ | $17.5 \pm 1.1$ |
| CoNLL: Organization | $57.8 \pm 1.6^{\dagger}$ | $21.2 \pm 1.3$ | Animal vs Object (3) | $48.4 \pm 3.1^{\dagger}$ | $27.3 \pm 2.8$ |
| Animal vs Object (5) | $36.3 \pm 3.0^{\dagger}$ | $16.3 \pm 2.3$ | Fruit vs Animal (3) | $45.1 \pm 3.1^{\dagger}$ | $21.5 \pm 2.5$ |
| Fruit vs Animal (5) | $28.1 \pm 2.8^{\dagger}$ | $9.9 \pm 1.9$ | Color vs Animal (3) | $63.3 \pm 3.0^{\dagger}$ | $18.6 \pm 2.4$ |
| Color vs Animal (5) | $47.0 \pm 3.1^{\dagger}$ | $9.5 \pm 1.8$ | Adjective vs Verb (3) | $57.3 \pm 3.1$ | $56.1 \pm 3.1$ |
| Adjective vs Verb (5) | $52.3 \pm 3.1^{\dagger}$ | $39.4 \pm 3.0$ | Verb vs Adjective (3) | $70.5 \pm 2.8^{\dagger}$ | $33.6 \pm 2.9$ |
| Verb vs Adjective (5) | $61.3 \pm 3.0^{\dagger}$ | $20.8 \pm 2.5$ | Concept vs Object (3) | $56.8 \pm 3.1^{\dagger}$ | $35.9 \pm 3.0$ |
| Concept vs Object (5) | $45.7 \pm 3.1^{\dagger}$ | $18.3 \pm 2.4$ | Object vs Concept (3) | $73.2 \pm 2.7^{\dagger}$ | $58.4 \pm 3.1$ |
| Object vs Concept (5) | $68.9 \pm 2.9^{\dagger}$ | $51.9 \pm 3.1$ | SQuAD Validation | $17.7 \pm 1.1^{\dagger}$ | $8.5 \pm 0.8$ |

Table A4: Performance comparison on **abstractive** in-context learning tasks (5-Shot) by ensuring that the target answer for each test instance is not included in the 5-shot examples. Values show accuracy $\pm$ 95% CI margin. Bold indicates higher performance; $\dagger$ denotes statistical significance ($p < 0.05$) according to McNemar's test.

| Task | Vanilla (%) | HAPAX (%) | Task | Vanilla (%) | HAPAX (%) |
|---|---|---|---|---|---|
| AG News | $0.3 \pm 0.3$ | $\mathbf{7.8 \pm 1.7}^{\dagger}$ | Antonym | $0.8 \pm 0.6$ | $\mathbf{2.5 \pm 1.0}^{\dagger}$ |
| Capitalize Second Letter | $0.5 \pm 0.5$ | $\mathbf{3.2 \pm 1.2}^{\dagger}$ | CommonsenseQA | $12.0 \pm 2.0$ | $\mathbf{13.9 \pm 2.1}$ |
| Country-Capital | $30.7 \pm 6.6$ | $\mathbf{42.3 \pm 7.0}^{\dagger}$ | Capitalize (Full Word) | $\mathbf{79.1 \pm 2.8}^{\dagger}$ | $62.2 \pm 3.3$ |
| Capitalize First Letter | $36.4 \pm 3.3$ | $\mathbf{73.6 \pm 3.0}^{\dagger}$ | Capitalize Last Letter | $2.9 \pm 1.2$ | $\mathbf{6.6 \pm 1.7}^{\dagger}$ |
| Country-Currency | $4.3 \pm 4.1$ | $4.3 \pm 4.1$ | Lowercase First Letter | $73.3 \pm 3.0$ | $\mathbf{76.2 \pm 2.9}$ |
| National Parks | $15.3 \pm 3.3$ | $\mathbf{22.6 \pm 3.9}^{\dagger}$ | Next Capital Letter | $3.4 \pm 1.3$ | $\mathbf{8.6 \pm 1.9}^{\dagger}$ |
| Landmark-Country | $30.4 \pm 3.1$ | $\mathbf{38.4 \pm 3.3}^{\dagger}$ | Lowercase Last Letter | $3.8 \pm 1.3$ | $\mathbf{10.0 \pm 2.1}^{\dagger}$ |
| Next Item | $12.0 \pm 4.2$ | $\mathbf{28.0 \pm 5.9}^{\dagger}$ | Park-Country | $10.7 \pm 2.2$ | $\mathbf{16.7 \pm 2.7}^{\dagger}$ |
| Present-Past | $54.6 \pm 5.7$ | $\mathbf{78.8 \pm 4.7}^{\dagger}$ | Previous Item | $5.3 \pm 2.9$ | $\mathbf{7.1 \pm 3.4}$ |
| Product-Company | $15.9 \pm 3.1$ | $\mathbf{24.9 \pm 3.7}^{\dagger}$ | Sentiment | $2.8 \pm 1.0$ | $\mathbf{35.4 \pm 3.0}^{\dagger}$ |
| Singular-Plural | $62.0 \pm 6.6$ | $\mathbf{77.1 \pm 5.8}^{\dagger}$ | Synonym | $1.6 \pm 0.8$ | $\mathbf{2.0 \pm 0.9}$ |
| Word Length | $7.7 \pm 1.8$ | $\mathbf{14.9 \pm 2.4}^{\dagger}$ | | | |

Table A5: Performance comparison on **abstractive** in-context learning tasks (5-Shot) for THRESHOLDED-HAPAX by ensuring that the target answer for each test instance is not included in the 5-shot examples. Values show accuracy $\pm$ 95% CI margin. Bold indicates higher performance; $\dagger$ denotes statistical significance ($p < 0.05$) according to McNemar's test.

| Task | Vanilla (%) | HAPAX THRESH (%) | Task | Vanilla (%) | HAPAX THRESH (%) |
|---|---|---|---|---|---|
| AG News | $0.3 \pm 0.3$ | $\mathbf{3.7 \pm 1.2}^{\dagger}$ | Antonym | $0.8 \pm 0.6$ | $\mathbf{7.0 \pm 1.6}^{\dagger}$ |
| Capitalize Second Letter | $0.5 \pm 0.5$ | $\mathbf{7.6 \pm 1.9}^{\dagger}$ | CommonsenseQA | $\mathbf{12.0 \pm 2.0}^{\dagger}$ | $7.3 \pm 1.6$ |
| Country-Capital | $\mathbf{30.7 \pm 6.6}^{\dagger}$ | $29.6 \pm 6.5$ | Capitalize (Full Word) | $\mathbf{79.1 \pm 2.8}^{\dagger}$ | $52.2 \pm 3.4$ |
| Capitalize First Letter | $\mathbf{36.4 \pm 3.3}^{\dagger}$ | $31.2 \pm 3.2$ | Capitalize Last Letter | $2.9 \pm 1.2$ | $\mathbf{8.2 \pm 1.9}^{\dagger}$ |
| Country-Currency | $\mathbf{4.3 \pm 4.1}^{\dagger}$ | $3.2 \pm 3.4$ | Lowercase First Letter | $\mathbf{73.3 \pm 3.0}^{\dagger}$ | $69.2 \pm 3.2$ |
| National Parks | $15.3 \pm 3.3$ | $\mathbf{25.3 \pm 4.0}^{\dagger}$ | Next Capital Letter | $3.4 \pm 1.3$ | $\mathbf{6.3 \pm 1.7}^{\dagger}$ |
| Landmark-Country | $\mathbf{30.4 \pm 3.1}^{\dagger}$ | $26.1 \pm 3.0$ | Lowercase Last Letter | $3.8 \pm 1.3$ | $\mathbf{5.2 \pm 1.5}^{\dagger}$ |
| Next Item | $12.0 \pm 4.2$ | $12.0 \pm 4.2$ | Park-Country | $10.7 \pm 2.2$ | $\mathbf{16.1 \pm 2.7}^{\dagger}$ |
| Present-Past | $\mathbf{54.6 \pm 5.7}^{\dagger}$ | $41.6 \pm 5.6$ | Previous Item | $\mathbf{5.3 \pm 2.9}^{\dagger}$ | $1.8 \pm 1.7$ |
| Product-Company | $15.9 \pm 3.1$ | $\mathbf{20.7 \pm 3.5}^{\dagger}$ | Sentiment | $\mathbf{2.8 \pm 1.0}^{\dagger}$ | $0.0 \pm 0.1$ |
| Singular-Plural | $\mathbf{62.0 \pm 6.6}^{\dagger}$ | $34.6 \pm 6.5$ | Synonym | $1.6 \pm 0.8$ | $\mathbf{3.8 \pm 1.2}^{\dagger}$ |
| Word Length | $\mathbf{7.7 \pm 1.8}^{\dagger}$ | $2.0 \pm 1.0$ | | | |

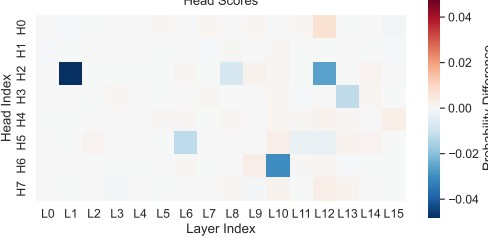

Figure A1: Probability difference values of random repetition task for Thresholded-HAPAX.

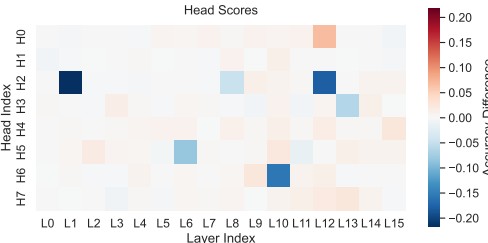

Figure A2: Accuracy difference values of random repetition task for Thresholded-HAPAX.

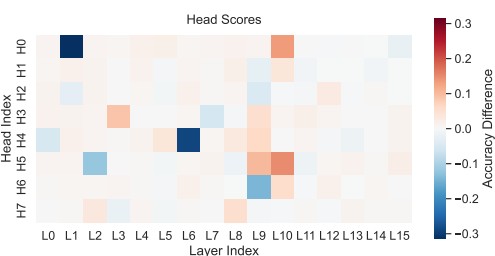

Figure A3: Accuracy difference values of random repetition task for HAPAX

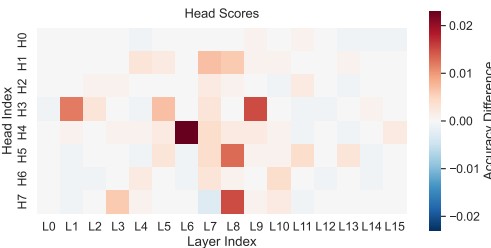

Figure A4: Accuracy difference values of random repetition task for vanilla model.

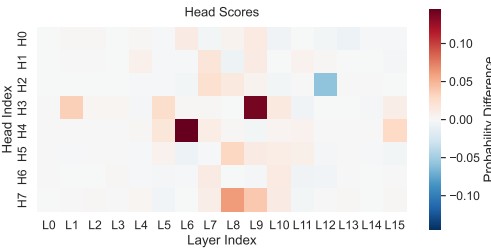

Figure A5: Probability difference values of random repetition task for vanilla model.

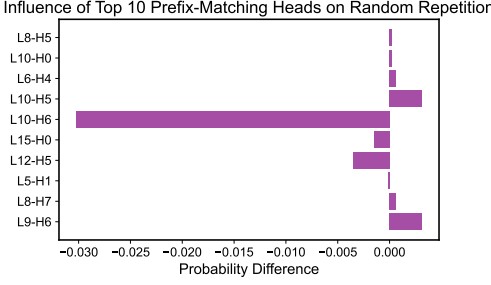

Figure A6: Probability Difference for random repetition task of the Top 10 Prefix-Matching heads for Thresholded-HAPAX.

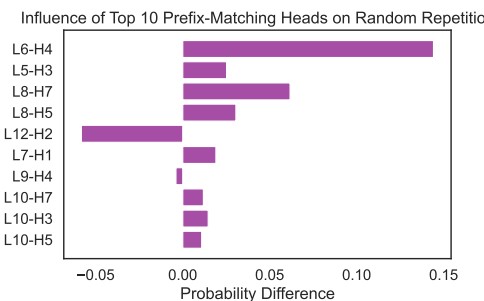

Figure A7: Probability Difference for random repetition task of the Top 10 Prefix-Matching heads for vanilla model.

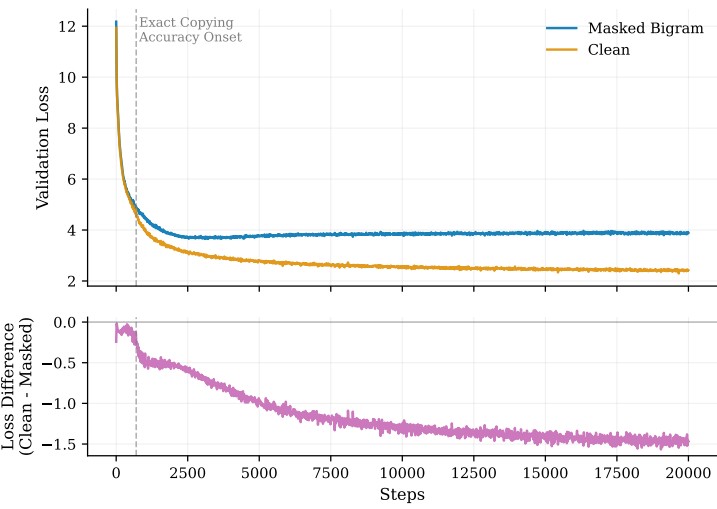

Figure A8: Validation Loss of Vanilla and Hapax model. Despite having increased loss value, as mentioned in the main paper, the validation loss also is mostly affected by the prediction of exact copying instances.

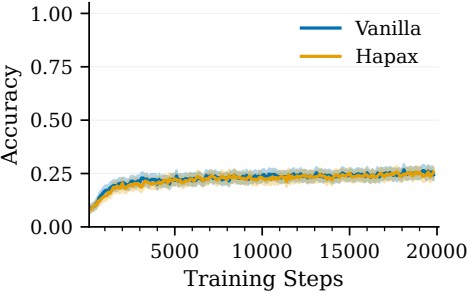

Figure A9: The non-repeating natural text repetition accuracy. Instead of giving a repeated sequence $r_1 r_2 ... r_s r_1 r_2 ... r_{s-1}$ we only give $r_1 r_2 ... r_{s-1}$. This shows that the initial increase seen in the natural text repetition also occurs for non-repeating samples.

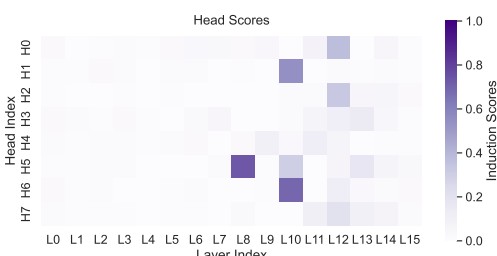

Figure A10: Prefix Matching Scores for Thresholded-HAPAX.

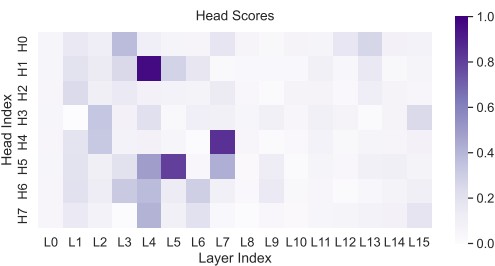

Figure A11: Previous Token Head of vanilla model at step 5000. L4 is the first layer that has a strong previous token head, which is L4 H1.

### A.3 ATTENTION PATTERN SCORES

## B COSINE SIMILARITY THRESHOLD SELECTION

The main purpose of Thresholded-HAPAX is to create a stricter training regime for suppression of inductive copying. Our Thresholded-HAPAX experiments are run to observe how well this masking reduces copying signal compared to both the regular model and the regular HAPAX model. In Figure 3a, the regular HAPAX model has highly reduced copying accuracy compared to the regular model but still at a non-trivial level. Our hypothesis is that even if we mask exact verbatim copying signals, there might still be token representations with high representational similarity, high enough to still provide a weak signal for verbatim copying. To test this, we select the cosine similarity threshold by using the embedding space of the last step of the vanilla model as a proxy for finding similar tokens. We create a cosine similarity matrix and analyze the cosine similarities of the closest neighbors and their edit distances. After doing this, we select the threshold of 0.3, which corresponds to the top-4 closest neighbors on average (Figure A12) and an average edit distance of 3.65 (Figure A13). Tokens representing different strings may still have highly similar embeddings, so our thresholding deliberately overestimates these hypothesized similar tokens, which makes us more confident that we are removing copying signal. Given our limited compute budget, we adopt this strict threshold to ensure that the trained model consistently masks the majority of similar tokens. Although this eventually causes harm for abstractive tasks when compared to the regular HAPAX model, we observe consistent gains for translation, suggesting that the model keeps capabilities for such instances despite the heavy masking. This control over copying also cannot solely be explained by undertraining, because the Thresholded-HAPAX model is near zero at copying while still achieving 29.6% accuracy on non-trivial tasks like Country-Capital. Therefore, even if it performs worse than the vanilla model, it serves as a proof of concept that the model can achieve non-trivial capabilities in abstractive tasks despite having even more reduced copying and prefix-matching scores.

To further justify our use of input embedding cosine similarity (instead of edit distance), we analyze the relationship between string-level edit distance and embedding cosine similarity for the top-50 nearest neighbors in the vanilla model's embedding space (Figure A15). With a cosine similarity threshold of 0.3, 7.24% of the token pairs are treated as equal, and only 1% of these pairs have an edit distance higher than 3. In contrast, if we instead threshold by edit distance at 3, 34.8% of the pairs are treated as equal, and 28.56% of these have a cosine similarity below 0.3. Moreover, an

edit distance threshold of 3 still fails to capture 1% of the pairs whose cosine similarity is above 0.3. Therefore, edit distance treats many more pairs as equal but still does not capture some high-cosine-similarity token pairs, which we hypothesize are more closely related to copying signal. Since attention heads attend by hidden state similarity, cosine similarity is closer to what an attention head would consider similar. We use cosine similarity as our base and use the average edit distance at different cosine similarity thresholds to guide us, so that we make sure we are removing tokens above a certain cosine similarity while also understanding how close they are on average to their represented tokens.

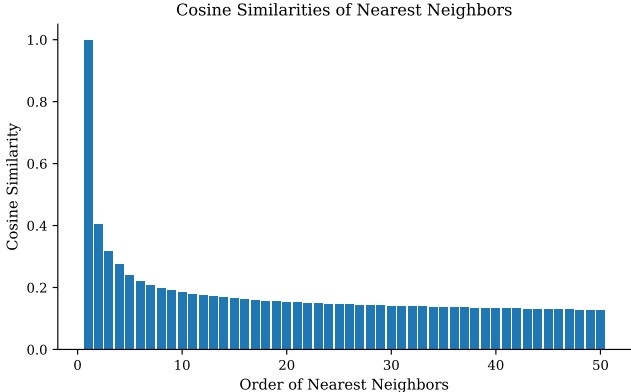

Figure A12: Cosine similarities of nearest N neighbors for clean model.

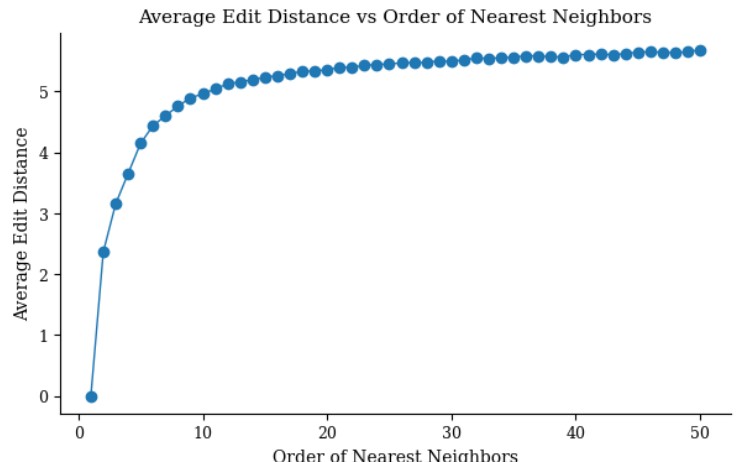

Figure A13: Average edit distance of nearest neighbors for vanilla model.

## C  HAPAX FLUENCY EVALUATION

We evaluate the effects of HAPAX training on the model's ability to generate fluent natural text and compare the results to its vanilla counterpart as a baseline. Inspired by the Fluency score from Axbench Wu et al. (2025), we use an automated LLM-based approach to assess the fluency of sentence completions generated by each model, with discrete ratings of 0, 1, and 2 for fluent language. Sequence continuations, with a maximum length of 128 tokens, are generated based on a diverse seed of 20 tokens, sampled from the RedPajama dataset Weber et al. (2024).

We find in Figure A16 that our HAPAX trained model generates somewhat fluent text 70% more times than the vanilla model. The latter, performing poorly, generates non-fluent language 84% of the time. Looking at some qualitative examples of generation in Figure A17, we observe that the

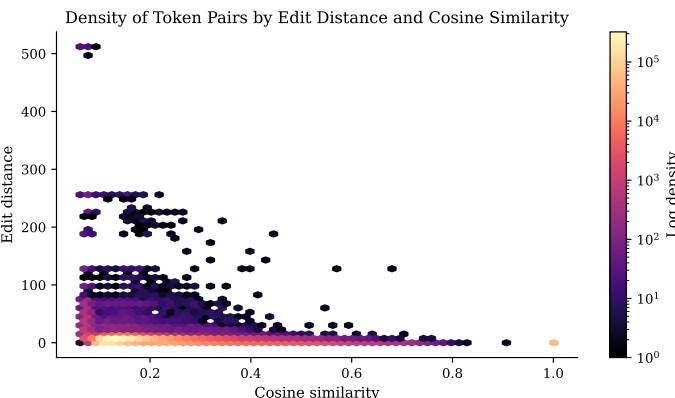

Figure A14: Low edit-distance pairs appear across a broad range of cosine similarities, suggesting that edit distance over-treats tokens as equivalent. The analysis is done on the same pairs analyzed in A13.

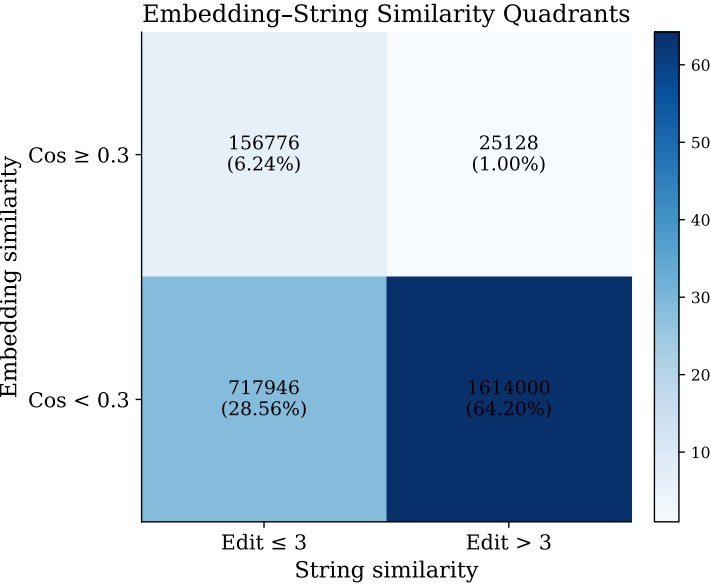

Figure A15: Relationship between string-level similarity (edit distance) and embedding similarity for the top-50 nearest neighbors. Many token pairs with low edit distance also have low embedding similarity (28.56%). However, 1% of the token pairs with cosine similarity above 0.3 are still not captured. Since attention heads attend by hidden state similarity, we want to make sure we are removing such similar tokens, which motivates our use of cosine similarity.

vanilla model defaults to repeating itself rather early, creating incoherent language, which explains its negative results. On the other hand, the HAPAX model is able to produce chained narratives with much more fluid and natural structure.

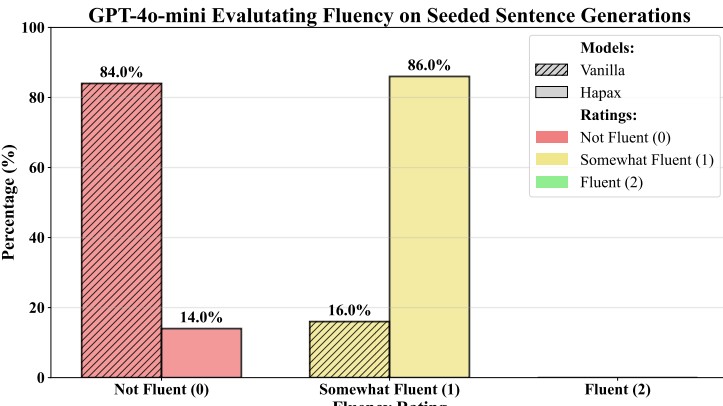

Figure A16: GPT-4o-mini as a judge rates 100 samples of natural text generated by each model based on fluency.

## D  EXPERIMENTAL DETAILS

### D.1  TRANSLATION TASKS

To perform translation evaluation, we constructed a multilingual translation dataset using the Open Multilingual Wordnet (OMW) 1.4 corpus via NLTK (Bird & Loper, 2004). To select languages, we used W3Techs[1] and chose languages that are both present in OMW 1.4 and have above 1% usage on the internet according to W3Techs. Since W3Techs data update dynamically, our selection was based on June 2025 data. Our choices resulted in 981 parallel concepts to be used for our tasks. We use accuracy as our evaluation metric. For each synset, we use only the first lemma from the source language to predict the corresponding English translation. All evaluations are 5-shot, and we translate from each source language into English to ensure consistent token counts across all tasks. An example prompt is shown in Figure A18.

## E  EXPLORED TRAINING VARIANTS

Before finalizing the HAPAX training protocol, we explored several variants to understand the impact of inductive copying on learning abstractive ICL capabilities. The methods proposed here focus mainly on the first part of the induction circuit, namely the prefix-matching behavior. The motivation behind these methods is to eliminate the initial circuit component, which we expected would remove inductive copying behavior completely. We summarize these variants and discuss why they fail and are therefore not suitable for our purposes.

### E.1  LOSS PENALTY BASED INTERVENTIONS FOR PREFIX-MATCHING SCORE

To analyze the impact of inductive copying on the development of abstractive ICL capabilities, we included a loss function that penalizes the prefix-matching attention pattern commonly observed in induction heads. Given our loss function $\mathcal{L}$, we add the prefix-matching score across all heads to our loss function:

$$\mathcal{L}_{\text{pm}} = \lambda \frac{1}{LH} \sum_{l=1}^{L} \sum_{h=1}^{H} \text{PrefixMatching}(l, h), \quad (5)$$

---

[1]https://w3techs.com/technologies/overview/content_language

**Prompt:**

Are you a resident of Pinnacle who owns a small business and operates from your home?

**Vanilla:**

If you answered "yes" to the question, you are a resident of Pinnacle.

If you answered "no" to the question, you are a resident of Pinnacle.

If you answered "yes" to the question, you are a resident of Pinnacle.

If you answered "no" to the question, you are a resident of Pinnacle…

**Hapax:**

If so, you are in luck.
The P.A.C.P.E. (Professional Association of Convenience Store Owners) is a non-profit organization that provides a wide range of services to the convenience store industry. The PCE is the largest membership organization in the PTA. It is dedicated to providing a safe and secure environment for the members of the convenience store and food service industry, and to promoting the health…

**Prompt:**

Jean (Jeannie) passed away after a lengthy battle with Emphysema and Diabetes. She

**Vanilla:**

was a beloved member of the St. Louis St. Louis St. Louis St. Louis St. Louis St. Louis St. Louis St. Louis St. Louis St. Louis St. Louis St. Louis St. Louis St. Louis St. Louis St. Louis St. Louis St. Louis St. Louis St. Louis St. Louis St. Louis St. Louis St. Louis St. Louis St. Louis St. Louis St. Louis St. Louis St. Louis St. Louis St. Louis St. Louis St. Louis St. Louis St. Louis St.

**Hapax:**

was born in New York City and raised in the Bronx.

She was a member of the New England Patriots, the Boston Red Sox, and the Boston Celtics. Her favorite sport was basketball. In her youth, she was an avid fan of her favorite team, Boston Celtics, which she played for in high school. When she retired from playing, her daughter, Jennifer, took over the care…

Figure A17: Examples of sentence completions generated for the evaluation. Each prompt seed is used to generate completions by both models. In both examples, the vanilla model's generation was rated with 0 (Not Fluent), while HAPAX received 1 (Somewhat Fluent).

```
"boite de conserve" - "can"
"adresse" - "address"
"abus" - "maltreatment"
"bétail" - "livestock"
"argile" - "mud"
"chose" - "
```

Figure A18: An example French-English translation. The prediction is accepted to be true if the next generation starts with *object"*, which includes both the correct generation *object* and the closing quotation mark *"*.

where $\lambda$ is our hyperparameter for fine-tuning. We propose two different versions. In the first one, we compute the prefix-matching score on the synthetically generated random repetition sequence. The random sequences are given in separate forward passes along with regular training. In the second version, instead of using synthetic data, we compute the prefix-matching score on the training data itself. We use the same positions as we use to mask the loss in Section 3.1.

The initial approach overfits on the synthetic data and consequently, the prefix-matching pattern and copying is intact for natural language. The second approach reduces prefix-matching score for both synthetic and natural settings. However, although we observe that the loss function can control the prefix-matching score, the model can still do random copying. We give random repetition accuracies for two different lambda values in Figure A19 for the non-synthetic variation. When we analyze individual attention patterns on the random repetition task, we find that the model learns to copy from two-tokens ahead, instead of displaying the regular prefix-matching attention pattern. As an example, in Figure A20, red squares indicate the regular prefix-matching positions. However, when we penalize the attention pattern, the model can still read the information from the 1-offset positions. Therefore, attention-pattern based training interventions are ill-suited for our purposes. This observation also motivate us to focus on *incentives*, as we did for HAPAX, rather than directly optimizing for a specific behavior, since the model's solution space tends to discover alternative ways to achieve the objective, similar to reward hacking.

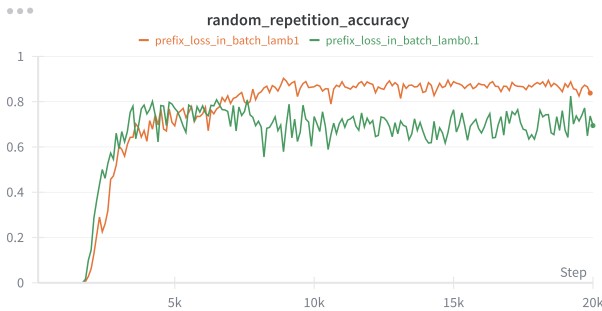

Figure A19: Random repetition accuracy for models trained with prefix-matching loss. Two lambda values that reduces the prefix-matching score are still able to learn random repetition.

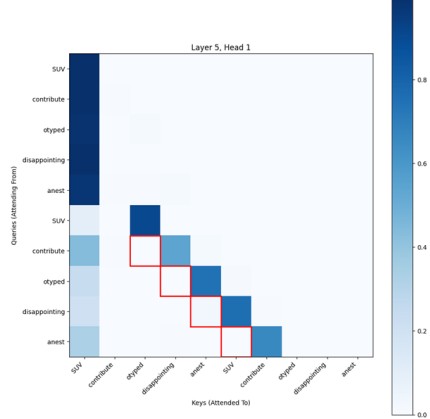

Figure A20: Example attention pattern of a model trained with a loss function that penalizes prefix-matching scores. Red squares indicate prefix-matching positions. Although the model does not display the typical attention pattern for induction, it learns to read the copying information from 1-offset positions.

## E.2 REINITIALIZING INDUCTION HEADS

To analyze whether weight reinitialization would allow us to suppress inductive copying, we have created two different training protocols. First, we reinitialized any attention head whose prefix-matching score was above a threshold $\tau$, which is 0.3 in our case. Secondly, in addition to reinitializing the weights, we have reinitialized the optimizer states for the corresponding attention heads that go above the threshold. For the first case, we have observed that the model can continue its training but the reinitialization creates more induction heads as compared to what it would learn in the vanilla version (Figure A22). We hypothesize that this might be due to the fact that ADAM optimizer builds up enough momentum which causes different heads to converge into induction heads as the training goes on. When we also reinitialize the corresponding optimizer states, the model training procedure gets stuck (Figure A21, A23) and causes every attention head to have an increased prefix-matching score. Therefore, when we keep the same data distribution, the model tends to converge on similar solutions for copying behavior, which makes the presented methods unsuitable for our purposes.

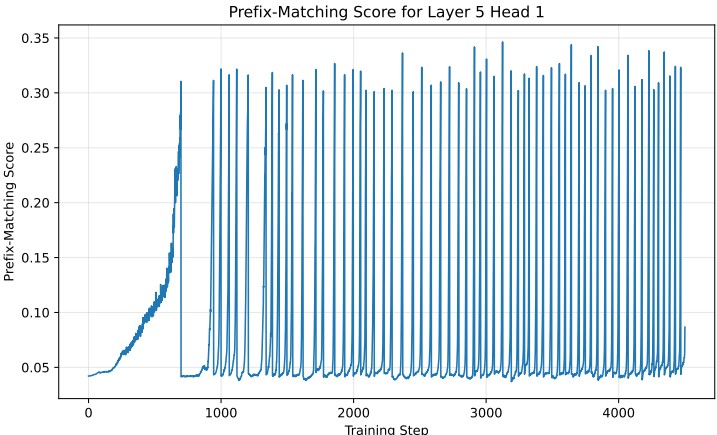

Figure A21: When the optimizer state for induction heads is reinitialized in combination with the weights, the model gets stuck and tries to relearn induction heads. We plot prefix-matching scores before we reinitialize, which is why certain data points exceed 0.3.

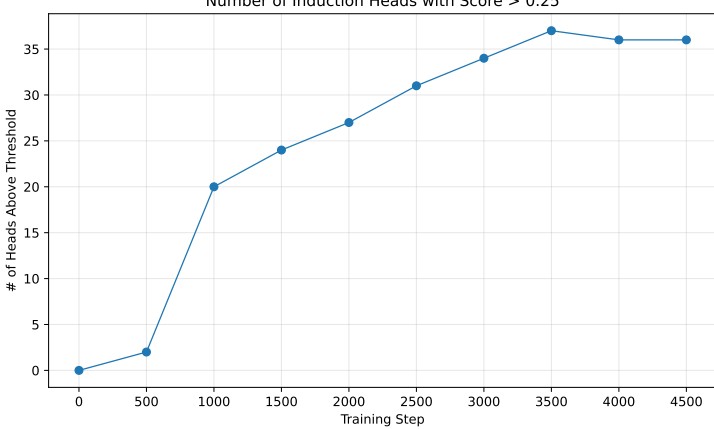

Figure A22: Number of induction heads that exceed the 0.25 threshold for the reinitalized model (160M). Since we reinitialize heads that exceed 0.3, we plot heads that exceed 0.25. The reinitialization causes the model to create three times more induction heads as compared to its vanilla version.

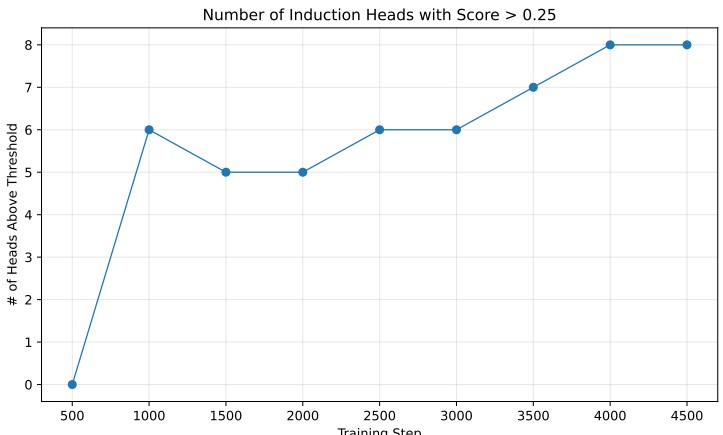

Figure A23: Number of attention heads that exceed 0.25 threshold for a vanilla model (160M).

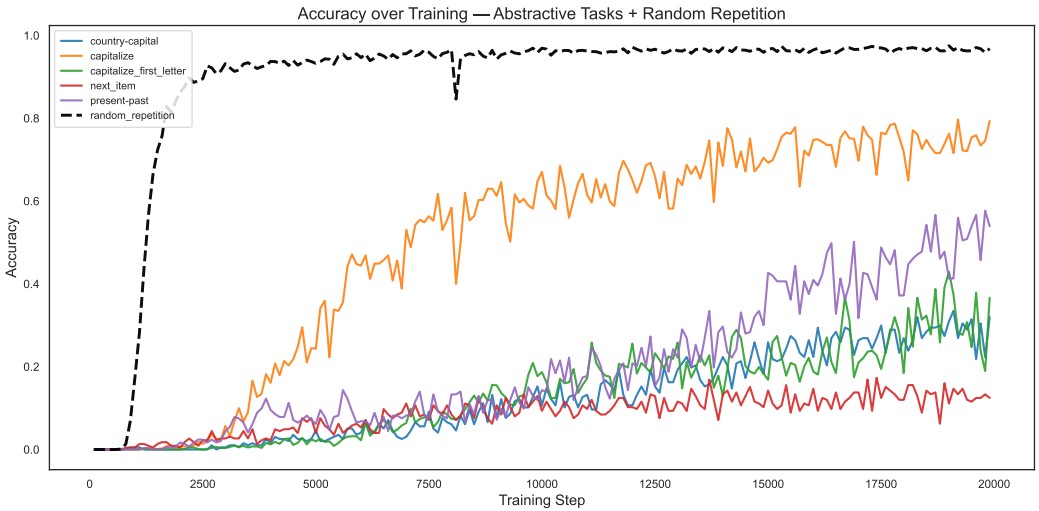

Figure A24: Accuracies of sample tasks over training for the Vanilla model. Random repetition accuracy is the first task to improve, and its rise coincides with the training step where the loss drops sharply (A8). This motivates examining whether the decrease in loss and the increase in inductive copying translate into meaningful gains on abstractive tasks, which according to our evidence are not tightly coupled.

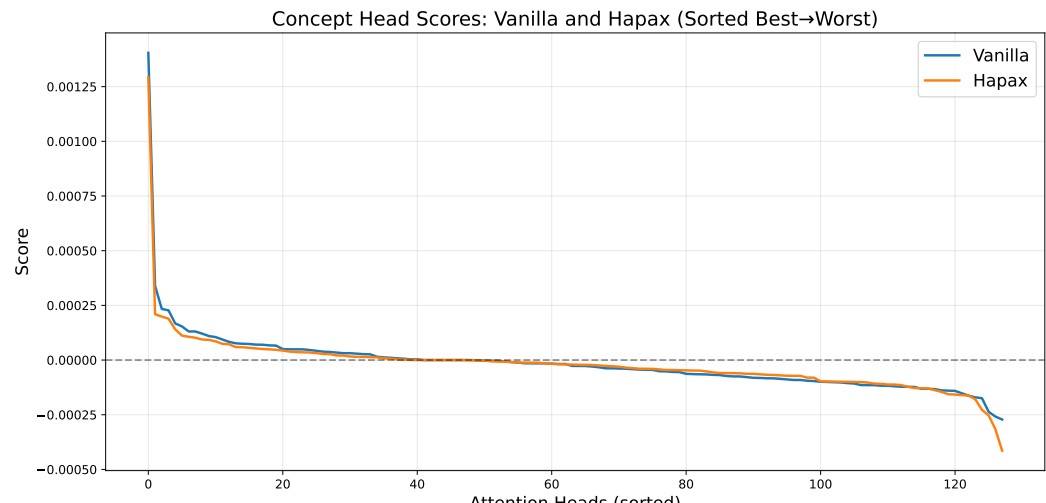

Figure A25: Using the concept head scores from Feucht et al. (2025), we display the scores of the models from best to worst. The trends are similar for both the Vanilla and HAPAX models, indicating that suppressing inductive copying does not noticeably affect the causal concept head metrics.

## F    USAGE OF LLMS

In accordance with the ICLR 2026 author guidelines, we made limited use of LLM-based services such as Claude and Grammarly as assistive tools for correcting grammar and refine writing.

## G    CHOICE OF STATISTICAL TESTING

For downstream tasks where evaluation reduces to accuracy, each example yields a paired binary outcome across models (correct vs. incorrect). McNemar's test is appropriate for this paired binary setting, as it assesses whether disagreements between the two models on the same examples are symmetric. Prior work has shown that McNemar's test is well-calibrated for classifier comparison in such conditions Dietterich (1998). Accordingly, we use McNemar's test for our main accuracy-based comparisons.

