# OpenReview forum: "In-Context Learning Without Copying"
_ICLR.cc/2026/Conference — Submitted to ICLR 2026_

### Official Review · Reviewer_qNXr · 2025-10-24

**Soundness:** 1
**Presentation:** 3
**Contribution:** 1
**Rating:** 0
**Confidence:** 3

**Summary:**

The authors study a scheme to discourage induction head formation while training a Transformer model. The authors assert that training with this scheme may benefit (or at least, not harm) a model's ability to perform well on "abstractive" ICL tasks.

**Strengths:**

The authors study an interesting topic, and forward a mildly provocative proposal that induction heads are unnecessary for ("abstractive") ICL.

**Weaknesses:**

It's unclear to what extent the authors' experiments support their assertions. Their statistical methodology is unusual to me, and their mechanistic claims are vague. In its present state, I am skeptical of their results. Please see Questions below for more details.

Even supposing their results are sound, I'm unsure how to judge the significance of their work. While it's a curious fact that suppressing copying during training can nonetheless leave some ICL capabilities intact, it appears that the large majority of ICL tests do suffer (looking at tables 1, A1, and A2). Will applying this loss scheme boost downstream performance in language modeling, logical reasoning, agentic applications, etc.? Is there a broader application for these results? If not, do these results shed more light on how Transformers operate? Your mechanistic analysis seems to focus on detecting whether there are copying heads remaining in your Hapax Transformer, rather than investigating how the Hapax Transformer can perform ICL tasks at all. If the primary goal is to suggest that copying is inessential to ICL, do you have a better understanding of how ICL can be implemented in the absence of copying?

**Questions:**

- Simultaneous hypothesis testing typically requires stricter differences / broader confidence intervals to judge statistical significance (see e.g. https://en.wikipedia.org/wiki/Multiple_comparisons_problem). For example, if you run 20 simultaneous tests, chances are you will have 1 statistically significant result at p = 0.05 even if the underlying differences are not significant. Did you take into account simultaneous testing when summarizing your results?
-  Assuming all measures of statistical significance are genuine, Table 1 seems to indicate that only 9 out of 23 "abstractive" ICL tests are significantly better when using Hapax (and 6 are significantly worse). I'm unsure if this supports your assertion that Hapax is better for abstractive ICL?
- Over how many random seeds are your CI's generated from?
- I'm unfamiliar with McNemar's test, but a quick Google search seems to suggest that McNemar's test is used for paired 2 x 2 contingency tables (https://en.wikipedia.org/wiki/McNemar%27s_test). Are your data interpretable as paired 2 x 2 contingency tables? Why not use a traditional t-test?
- Section 4.3: I'm unsure what the takeaway is here. You appear to be claiming that ICL performance remains strong in the Hapax model using this token loss metric. However, you also acknowledge on Line 308 that this metric does not correlate well with ICL performance. So why use it at all?
- Section 5: also unsure what the takeaway is here. Why not study mechanistically why Hapax performs well on the abstractive subset of ICL tasks? It's quite plausible that your loss formulation suppresses copying (Figure 3 is quite convincing already), so I'm unsure why you need a mechanistic analysis of whether Hapax is forming induction heads?
- Figure 8: it's unclear to me whether these induction head scores are above chance. You seem to suggest on Line 431 that 0.10 thresholds the top 5 prefix matching heads from the vanilla model in Figure 6a. On inspection, however, the top 5 in Fig. 6a seem to be thresholded by ~0.9. A score of 0.10 seems to be close to the background chance level, based on your color scale? The color scale on Figure 8 seems to exaggerate the strength of the prefix scores in your random model?

---

> ### Author Response · Authors · 2025-11-26
>
> We thank the reviewer for their detailed comments and suggestions.
>
> **Clarity on how the paper sheds light on how transformers operate**: Several mechanisms beyond induction heads have been proposed to explain how transformers perform in-context learning, such as function vector heads and concept heads. However, prior discussions have hypothesized a dependence where these heads emerge from induction heads, (Olsson et al. (2022), Yin et al. (2025)), so the causal relationship was unclear. Our paper takes an initial step to address this gap by directly analyzing this causal relationship and gives insight into the training dynamics of how such mechanisms develop. Our paper examines this question directly by weakening the model’s inductive copying ability and consequently induction circuits. We observe that non-copying mechanisms remain present and effective. This provides evidence that these mechanisms do not rely strongly on each other in order to form. In this sense, our work sheds light on a key training dynamics question: whether learning abstractive behavior depends strongly on induction. Our results show that abstractive capability remains intact even when copying is suppressed and induction heads are substantially weakened. In addition, HAPAX omits loss on 31.7% of training token positions while preserving abstractive performance, suggesting that suppressing redundant repetition signals may inform future work on more efficient training approaches.
>
>
>   The reviewer also asks why we focus our analysis on the detection of copying circuits. This is because, to our knowledge, we are among the first to intervene in the training process of an LLM (instead of toy or synthetic models), with the goal of suppressing inductive copying and observing the resulting effects on its abstractive capabilities. Our intervention is designed to discourage the formation of inductive copying, therefore a central purpose of our investigation is to check whether our intervention had the intended effects. Beyond that, we find it informative that there is still a non-zero degree of copying inside the model despite never seeing repeated n-grams and analyzing possible reasons for how this might emerge, as we do in Section 5.
>
> **Performance of HAPAX**: We thank the reviewer for this point. Our central claim is not that Hapax universally improves abstractive ICL mechanisms, but that learning abstractive capabilities does not depend on copying and induction. Showing that Hapax maintains comparable performance on abstractive tasks supports this claim. Our statements about higher accuracy simply reflect that more tasks show improvement than decline. This does not imply that the model is uniformly superior for all abstractive mechanisms. The improvements we observe on a subset of tasks are encouraging but remain secondary to our main argument. In line with the reviewer's concerns, we have revised parts of the tone and wording in the paper to emphasize preservation of capabilities and improvements on specific tasks.
>
> That said, for abstractive tasks, we have added a second evaluation setting that controls for small label spaces, where the target token frequently appears in the 5-shot context. To prevent the vanilla model from benefiting through label-distribution matching, we reran the abstractive evaluation using only few-shot examples that exclude the target token. We observed that some of the vanilla model’s apparent advantages were due to distributional copying of the few-shot labels (for example, outputting one of the example labels at random). Under this control, Hapax achieves higher accuracy than the vanilla model on 17 out of 18 significant tasks for Table A4. Thresholded-Hapax also improves relative to the previous evaluation, achieving higher accuracy on 10 out of 22 significant tasks for Table A5. These findings suggest that part of the vanilla model’s earlier advantage came from distributional copying of the few-shot labels. We also note that these occur despite Hapax and Thresholded-Hapax being trained on only 31.7% and 52.5% of tokens, respectively. However, since our primary focus is on understanding training dynamics, we agree with the reviewer that the paper benefits from presenting these results with more neutral observations and have adjusted accordingly.

---

> ### Author Response · Authors · 2025-11-26
>
> **Simultaneous Hypothesis Testing**:
> Thank you for raising this point. Our analysis reported per-task significances without applying a multiple-hypothesis correction. While our scientific focus is on preservation of abstractive mechanisms rather than demonstrating global superiority, we agree that correction is appropriate given the number of simultaneous tests in Table 1. In response, we have applied the Benjamini–Hochberg correction. The same set of tasks remain statistically significant after correction. We have also revised the surrounding text to emphasize preservation of abstractive mechanisms. We are happy to make further adjustments if helpful.
>
> | Task                     |       Raw p | Benjamin-Corrected p|
> |--------------------------|-------------|-------------|
> | ag_news                  | 6.60827e-45 | 7.59951e-44 |
> | antonym                  | 0.0176221   | 0.0270205   |
> | capitalize               | 1.16801e-19 | 6.71608e-19 |
> | capitalize_first_letter  | 9.66767e-38 | 7.41188e-37 |
> | capitalize_last_letter   | 5.70451e-07 | 1.64005e-06 |
> | capitalize_second_letter | 8.2615e-12  | 3.16691e-11 |
> | commonsense_qa           | 6.91142e-16 | 3.17925e-15 |
> | country-capital          | 0.00298491  | 0.00492447  |
> | country-currency         | 0.877371    | 0.917251    |
> | landmark-country         | 0.00160098  | 0.00306854  |
> | lowercase_first_letter   | 0.148979    | 0.201559    |
> | lowercase_last_letter    | 0.75183     | 0.823432    |
> | national_parks           | 0.0509619   | 0.0732578   |
> | next_capital_letter      | 0.609834    | 0.701309    |
> | next_item                | 4.63085e-07 | 1.52157e-06 |
> | park-country             | 0.0029975   | 0.00492447  |
> | present-past             | 6.99907e-07 | 1.78865e-06 |
> | prev_item                | 0.359283    | 0.434922    |
> | product-company          | 0.926652    | 0.926652    |
> | sentiment                | 1.2974e-75  | 2.98402e-74 |
> | singular-plural          | 0.00114318  | 0.00239028  |
> | synonym                  | 0.000258806 | 0.000595253 |
> | word_length              | 0.329114    | 0.420535    |
>
>
> **CI & Seeds**:  The confidence intervals we report are binomial normal-approximation intervals computed over test samples for a single trained model. They are not based on variability across multiple training seeds. Because our models have 1B parameters, we are constrained in how many full retrainings we can run and therefore do not perform multi-seed retraining. We do not use these intervals to claim statistical significance between models, they are included only to illustrate test-time variability for the given model.
>
> **McNemar’s Test**: For downstream tasks where evaluation is done by accuracy, we treat each example as producing a paired binary outcome (correct or incorrect) across models. McNemar’s test is designed for this paired binary setting, as it evaluates whether disagreements between two models on the same examples are symmetric. Prior work has shown that McNemar’s test is well-calibrated for classifier comparison in this setting (Dietterich, 1998 [https://sci2s.ugr.es/keel/pdf/algorithm/articulo/dietterich1998.pdf]). We therefore use McNemar’s test for our main accuracy-based comparisons. We have revised the paper to include our choice statistical testing in Appendix G.
>
> **Statistical Significance of Figure 8**: We apply a paired t-test to assess whether each attention head’s prefix-matching score differs from the mean score of all other heads across the 1000 samples. To focus on meaningful effects, we use our threshold to highlight top heads that are also the top prefix-matching heads in the clean model, rather than those that are merely statistically different. Because patching is performed on a randomly initialized model, even small deviations become statistically significant, causing many heads to pass significance. However, the heads with the highest prefix-matching scores consistently align with those in the fully trained model, which motivates our threshold choice.
>
> **Token-Loss Metric**: We use token-loss difference because it is central to Olsson et al. (2022)’s argument that induction heads drive in-context learning. Our contribution is to adapt this metric to make it more informative in an abstractive setting. As we note in Section 4.3, the original formulation is particularly sensitive to examples solvable through inductive copying. To address this, we introduce an adjusted metric (Figures 5a and 5c) that excludes such instances. When these cases are removed, the Hapax model’s ICL capabilities remain unchanged for tokens that cannot be predicted via copying. In contrast, the traditional metric is heavily influenced by exact-copying examples, which can obscure the behavior we aim to analyze. Our adjusted metric thus provides a complementary perspective and can be used to track abstractive capabilities.

---

> ### Comment · Reviewer_qNXr · 2025-11-26
>
> Thanks for the additional clarifying comments and elaboration.
>
> Ultimately, after reading your responses and that of the other reviewers, I remain in agreement that it's unclear how substantive the contribution is. Thank you for repositioning the paper to argue that a Transformer trained with HAPAX can nonetheless continue to perform ICL, rather than that HAPAX exceeds the performance of a vanilla Transformer. All the same, as reviewer 3Arf points out, it's already acknowledged that induction heads are not solely responsible for ICL. What additional do we gain from this analysis? If it's already known that ICL depends on many skills beyond simple copying, is it surprising that suppressing copying will nonetheless leave ICL capability intact?
>
> Thank you for the additional details on your hypothesis testing. Note, most hypothesis tests assume independent and identically distributed data. I'm unsure whether reusing the same trained model would constitute iid samples, when the intervention you're testing relies on a change in the training procedure, since each draw would be correlated with other draws sharing the same model. Additionally, I believe a BH correction also relies on independent tests, though it doesn't sound like your tests would truly be independent if the underlying model is the same. A Bonferroni correction may be more appropriate in this case (https://en.wikipedia.org/wiki/Bonferroni_correction).

---

> ### Author Response · Authors · 2025-11-28
>
> Thank you for the follow-up. A central motivation for our work comes from how Olsson et al. framed induction heads as a major component of ICL (*In-Context Learning and Induction Heads*, Introduction):
>
> > “induction heads might constitute the mechanism for the actual majority of all in-context learning in large transformer models.”
>
> Subsequent work has shown that induction circuits are not important at the end of training for different mechanisms involving non-copying, when more semantically grounded heads dominate (Feucht et al., Yin et al.). However, this still leaves an open question: Do induction heads act as early precursors that enable other ICL mechanisms to form?
>
> Yin et al. conjecture that induction heads are an early version of later FV heads that are gradually superseded during training (*Which Attention Heads Matter Most for In-Context Learning*, Section 6):
>
> > “Our first conjecture (C1) posits that induction heads are an early version of FV heads.”
>
> Motivated by this, our work tests whether induction heads are a necessary prerequisite. We suppress inductive copying and weaken induction heads from the outset, and find that abstractive ICL capabilities are preserved, despite a strong reduction in inductive copying and induction mechanisms, suggesting that the emergence of non-copying mechanisms does not necessarily depend on induction as a precursor.
>
> ---
>
> **References**
>
> [1] Olsson, et al. *In-context Learning and Induction Heads*, Transformer Circuits Thread, 2022.
>
> [2] Yin, K., & Steinhardt, J. *Which Attention Heads Matter for In-Context Learning?* OpenReview, 2025. https://openreview.net/forum?id=KadOFOsUpQ
>
> [3] Feucht, S., Todd, E., Wallace, B. C., & Bau, D. *The Dual-Route Model of Induction*. Second Conference on Language Modeling, 2025. https://openreview.net/forum?id=bNTrKqqnG9
>
> As of Dec 2025, Google Scholar citation counts: [1] 711; [2] 23; [3] 7.

---

### Official Review · Reviewer_r34s · 2025-10-29

**Soundness:** 2
**Presentation:** 3
**Contribution:** 2
**Rating:** 2
**Confidence:** 3

**Summary:**

The authors train models with a modified next-token loss, which aims to mask repetitions in order to investigate ICL in models that are disincentivized to form induction heads.

**Strengths:**

- The authors introduce a novel framework to investigate ICL in a setting where induction head formation is discouraged
- I believe the question of what transformers can learn when they are disallowed from forming induction heads is interesting and may lead to the discovery of important transformer circuits
- The Hapax scheme the authors propose is an interesting lens into the formation of induction heads and what signals are required for such circuits to form, I encourage the authors to continue with this work

**Weaknesses:**

- It is not clear to me that input embedding similarity is the right way to resolve repetitions in text that are tokenized differently.
    - Tokens may have similar embeddings despite representing distinct strings
    - It is not clear to me (even after reading appendix B) that the threshold chosen by the authors ($\tau = 0.3$) effectively suppresses repetitions in natural text.
        - I cannot find experiments testing the effect of $\tau\not=0.3$, or experiments that show that embedding cosine similarity is an effective measure of token string similarity (For one thing, Figure A11 shows only the *average* edit distance, and not the full distribution, nor any notion of concentration around the mean)
- I am not sure I agree with the claim that Hapax performs better on abstractive ICL tasks.
    - In table 1, the vanilla model still performs better on a significant fraction of the tasks, and although the authors do perform a statistical analysis that analyzes the accuracy distribution of each model trained, the results are mixed enough to call into question whether a different vanilla model trained with a different random seed might perform better than Hapax on many of the tasks here.
        - I.e. the authors characterize statistical fluctuations only over the performance of one model, not over the distribution of models -- while I recognize that characterizing the full distribution is probably computationally prohibitive, the results for a single model are inconclusive enough that I do not feel comfortable with the statement "Hapax improves abstractive ICL"
    - The authors also note that thresholded Hapax performs worse than the vanilla model: this could be because of an improperly chosen threshold (see my earlier point), or because the distribution over models trained via Hapax is sufficiently noisy as to produce inconsistent results. In either case, I believe further investigation is necessary.

**Questions:**

- Can the authors provide more concrete evidence that $\tau=0.3$ is a good choice of threshold? Both in the sense of downstream abstractive ICL performance, and with respect to masking duplicate tokens without masking non-duplicate tokens that may have significant embedding similarity?
    - The fact that thresholded Hapax with $\tau=0.3$ shows poor performance on abstractive ICL tasks calls into question whether the threshold was correctly chosen.
- Can the authors make precise their claim that Hapax performs better on abstractive ICL tasks? Currently the results are noisy enough that I am not sure that this claim would hold if one were to retrain Hapax models with different seeds.

---

> ### Author Response · Authors · 2025-11-26
>
> We thank the reviewer for their thoughtful and encouraging feedback and appreciate their recognition of our framework and the HAPAX scheme.
>
>
> **Choice of Threshold**: Our Thresholded-HAPAX experiments were run to observe how well our masking reduces copying signal. In Figure 3a, the regular HAPAX model has highly reduced copying accuracy compared to the regular model but still at a non-trivial level. Our hypothesis was that even if we mask exact verbatim copying signals, there might still be token representations with high representational similarity, high enough to still provide a weak signal for verbatim copying. As the reviewer accurately observes, tokens representing different strings may still have highly similar embeddings. Our thresholding deliberately overestimates these hypothesized similar tokens, which makes us more confident that we are removing copying signal. We agree that a more fine-grained similarity selection could give better results, but our intention with Thresholded-HAPAX was to show that we can control copying reliably with our masking. This control also cannot solely be explained by undertraining, because the Thresholded-HAPAX model is near zero at copying while still achieving 29.6% accuracy on non-trivial tasks like Country-Capital (even when trained on ~52% less tokens).
>
>
> Also, as attention heads attend by hidden state similarity, cosine similarity is closer to what an attention head would also consider similar. We used cosine similarity as our base and used the average edit distance at different cosine similarity thresholds to guide us, so that we make sure we are removing tokens above a certain cosine similarity while also understanding how close they are on average to their represented tokens. We have added Figure A15 to illustrate this point, and added Figure A14 for a more informative visualization of the distribution. If we consider the top 50 nearest neighbors for each token as pairs, with a cosine similarity threshold of 0.3, 7.24% of the pairs are treated as equal, where only 1% of them have a edit distance higher than 3. If we instead choose an edit distance of 3, 34.8% of the pairs are treated as equal, and 28.56% of these pairs have a cosine similarity below 0.3. Moreover, even with an edit distance of 3, 1% of the pairs that have cosine similarity above 0.3 are not counted as equal. Therefore, edit distance treats many more pairs as equal but still does not capture some high-cosine-similarity token pairs, which we hypothesize are more related to the signal of copying. Based on this analysis, we use cosine similarity as our thresholding basis.
>
>
> **Performance of HAPAX**: We thank the reviewer for this point. Our central claim is not that Hapax universally improves abstractive ICL, but that learning abstractive capabilities does not depend on copying and induction. Demonstrating that Hapax performs well enough on these tasks would be sufficient to support our point. The improvements we observe on a subset of tasks are encouraging but remain secondary to our main argument. In line with the reviewer's concerns, we have revised parts of the tone and wording in the paper to emphasize preservation of capabilities and improvements on specific tasks rather than implying a global improvement. If there are any other sections that could potentially be misinterpreted, we are happy to revise those as well.
>
> That said, for abstractive tasks, we have added a second evaluation setting that controls for small label spaces, where the target token frequently appears in the 5-shot context. To prevent the vanilla model from benefiting through label-distribution matching, we reran the abstractive evaluation using only few-shot examples that exclude the target token. We observed that some of the vanilla model’s apparent advantages were due to distributional copying of the few-shot labels (for example, outputting one of the example labels at random). Under this control, Hapax achieves higher accuracy than the vanilla model on 17 out of 18 significant tasks (Table A4). Thresholded-Hapax also improves relative to the previous evaluation, achieving higher accuracy on 10 out of 22 significant tasks (Table A5). These findings suggest that part of the vanilla model’s earlier advantage came from distributional copying of the few-shot labels. We also note that these occur despite Hapax and Thresholded-Hapax being trained on only 31.7% and 52.5% of tokens, respectively. However, since our primary focus is on understanding training dynamics, we agree with the reviewer that the paper benefits from presenting these results with more neutral observations and have adjusted accordingly.

---

### Official Review · Reviewer_FhuR · 2025-10-31

**Soundness:** 3
**Presentation:** 3
**Contribution:** 4
**Rating:** 8
**Confidence:** 3

**Summary:**

This paper revisits a central assumption in mechanistic interpretability: that induction heads, which perform inductive copying of token sequences, are the foundation of in-context learning (ICL) in transformers. The authors introduce HAPAX, a novel training regime that removes the loss contribution of any token that could be predicted by induction heads—that is, any repeated n-gram within a context window. This loss masking prevents the model from receiving gradient signals for repeated sequences, thereby suppressing the incentive to learn inductive copying. Despite a 31.7% reduction in effective training tokens, HAPAX models maintain and often improve performance on abstractive ICL tasks, while performing worse on purely extractive, copy-based tasks. Mechanistic analysis shows that HAPAX models develop fewer and weaker induction heads but preserve contextual reasoning and even generate more fluent text. These results demonstrate that inductive copying is not necessary for abstractive in-context learning.

**Strengths:**

The paper’s originality lies in directly challenging one of the most established causal hypotheses about ICL. Rather than building another interpretability tool, the authors use a simple yet powerful experimental manipulation to test whether transformers can still learn ICL without explicit copying. This approach transforms a long-standing correlational observation into a causal experiment. The outcome is both surprising and illuminating: models deprived of copying signals still learn and sometimes excel at abstract reasoning tasks. Conceptually, this is an important reframing of how context learning emerges in large language models.

The technical and methodological quality is high. The HAPAX regime is clearly defined, with precise mathematical formalization of the masked loss and a thoughtful extension to similarity-thresholded masking. The results are carefully interpreted: the authors separate extractive versus abstractive tasks, use token-loss difference metrics to analyze contextual dependence, and provide attention-level evidence that induction heads weaken under HAPAX training. The work is also clear and well-presented, with figures and tables that clearly communicate both behavior-level and mechanistic results. Its significance is substantial, as it undermines the simplistic “ICL = copying” view and encourages a richer understanding of transformer learning dynamics.

**Weaknesses:**

The scope of empirical validation is somewhat limited. Experiments use 1B-parameter GPT-NeoX models trained on The Pile, which is appropriate for controlled analysis but smaller than models where ICL phenomena are most pronounced. It remains unclear whether the same findings generalize to multi-billion-parameter transformers or to multi-layer SAEs and circuit configurations used in interpretability research. The masking strategy targets repeated n-grams, which suppresses literal copying but not necessarily semantic or structural repetition, meaning that some forms of inductive behavior may persist.

**Questions:**

1. To what extent would the results scale to larger models such as 7B or 13B transformers? Does suppressing inductive copying remain beneficial for abstractive ICL at higher capacities?
2. Since the paper argues that abstractive ICL arises independently of induction heads, have the authors identified any alternative head types or intermediate representations that correlate with abstractive task success?

---

> ### Author Response · Authors · 2025-11-26
>
> We thank the reviewer for their encouraging feedback and kind remarks. We truly appreciate your recognition and are glad to hear your enthusiasm for the paper.
>
> **Larger Parameter Size**: While resource limitations prevented us from running multi-billion-parameter experiments, we agree that extending the analysis to larger models is an important direction. We estimate that training two 6.9B (Vanilla and Hapax) models for 20K steps requires approximately 10,000 A100 GPU hours, corresponding to about $13K on AWS. We are currently exploring opportunities to scale our experiments further. Regarding whether abstractive tasks would improve under our approach for larger scale models, since we reduce repetitions, it is plausible that shifting the training distribution toward less repetitive inputs may benefit specific abstractive mechanisms by placing greater emphasis on the non-repetitive portion of the distribution.  Moreover, our central claim regarding the viability of abstractive ICL mechanisms under inductive-copying suppression should continue to hold at larger scales. As the reviewer notes, larger models generally exhibit stronger ICL and greater representational capacity, which is consistent with the expectation that suppressing inductive copying does not prevent the emergence of abstractive mechanisms at larger scale.
>
> **Semantic Repetition**: Our Thresholded-HAPAX masking can be viewed as a form of semantic repetition masking. Because we apply a cosine-similarity threshold, the masking suppresses both exact repetitions and semantically related ones, using input embedding similarity as a proxy. However, as the reviewer accurately notes, this approach still operates at the token level and may not fully address structural repetition involving multi-token words or phrase-level similarities. We agree that exploring higher-level masking criteria such as concept-level similarity or phrase-level masking could give even more precise control.
>
> **Correlation with Attention Heads**: We thank the reviewer for this question. Building on prior work, we examined attention heads associated with abstractive behavior in earlier mechanistic studies. In particular, we evaluated concept heads using the causal mediation–based concept copying score introduced in Feucht et al. (2025). We now have included these results in Figure A25. As shown in the figure, the relative ordering of concept copying scores is similar for both the Vanilla and Hapax models, indicating that suppressing inductive copying does not noticeably affect the emergence of concept heads. In prior work and in our experiments, the absolute magnitudes of these scores tend to be small, but their relative differences are meaningful and are used to compare heads. These observations are consistent with the view that concept-level mechanisms can develop in parallel when inductive copying signals are suppressed.

---

### Official Review · Reviewer_3Arf · 2025-11-01

**Soundness:** 2
**Presentation:** 2
**Contribution:** 1
**Rating:** 2
**Confidence:** 4

**Summary:**

This paper investigates whether **inductive copying**, often attributed to induction heads, is necessary for in-context learning (ICL).
The authors propose **HAPAX**, a training regime that masks the loss contribution of repeated n-grams within each context.
By excluding gradient signals from positions predictable through repetition, HAPAX aims to prevent the model from learning copy-based induction circuits.
Empirically, HAPAX models maintain—and sometimes improve—their performance on abstractive ICL tasks, while showing clear degradation on extractive or copy-heavy ones.
Mechanistic analysis indicates fewer and weaker prefix-matching heads, suggesting that inductive copying is not essential for abstractive ICL.

**Strengths:**

**Interesting empirical perspective.**
  The paper introduces a creative intervention—masking losses on repeated n-grams—to examine how removing copy-related learning signals affects in-context learning.
  This approach provides a valuable window into how specific training signals shape internal circuits, offering an original empirical angle rather than a purely conceptual contribution.

- **Clear experimental setup.**
  The masking mechanism and evaluation procedure are defined precisely and are easy to replicate.

- **Mechanistic interpretability link.**
  The analysis connects observed behavioral changes to head-level attention patterns, providing interpretable signals about circuit adaptation.

**Weaknesses:**

### 1. Conceptual framing may mislead rather than innovate
The central claim—that *inductive copying is not essential for ICL*—could be interpreted as overturning a previously dominant view that induction heads are the sole foundation of ICL.
In practice, the field already recognizes that induction heads support certain ICL behaviors but are not the only mechanism.
Thus, the framing could **unintentionally give the impression** that the paper refutes a consensus that did not exist.
The genuine novelty lies in the **empirical intervention itself**—examining how the removal of repetition-related gradients alters model behavior and circuit formation.
Reframing the work around this insight, rather than as a conceptual correction, would make the contribution clearer and more accurate.

### 2. Induction heads are reduced, not removed
The analysis shows that prefix-matching heads become fewer and weaker but not absent.
The framing could more explicitly emphasize this partial suppression to avoid readers inferring complete removal.

### 3. Ambiguity in how the masking rule affects learning
HAPAX masks all token positions where a bi-gram has appeared earlier in the same context.
This means only the first occurrence contributes to the loss.
While this design reduces gradients from repetition, it still exposes the model to repeated patterns and may unintentionally **encourage single-exposure learning**, where the model must internalize a pattern from its first appearance.
The results demonstrate that HAPAX changes how repeated patterns contribute to learning, but it remains unclear whether this constitutes a genuine reduction in memorization or simply a shift in its form.
Clarifying this distinction—between preventing repetition-based learning and altering how memorization occurs—would make the causal argument more precise.

### 4. The link between “copy suppression” and “induction suppression” is theoretically underspecified
Induction heads are not mere copy circuits—they implement a retrieval mechanism that associates similar contexts \(X_q\) and \(X\) to predict the continuation \(Y\).
Masking repeated n-grams blocks gradients for exact duplication but does not necessarily disrupt this broader retrieval mechanism.
Thus, while the intervention reduces surface-level repetition, it does not clearly isolate the mechanisms that drive generalization or memory formation.
A more detailed discussion of what aspects of induction behavior the masking targets (e.g., exact copying vs. contextual retrieval) would clarify interpretation.

**Questions:**

1. Does the mask exclude every *second and later* occurrence of a bi-gram globally, or only within a local window?
   Could this setup unintentionally promote single-exposure learning instead of preventing memorization?
2. How do you interpret the reduction in prefix-matching heads—does it reflect fewer copy circuits or simply less training on repeated n-grams?
3. Have you checked whether the model continues to exhibit *semantic* or *fuzzy* induction behaviors despite masking?
4. Would removing repeated n-grams entirely from the input, rather than from the loss, yield a clearer causal result?
5. In what sense do you consider HAPAX “without copying,” given that repetition remains visible to the model?

---

> ### Author Response · Authors · 2025-11-26
>
> We thank the reviewer for a thorough analysis of our paper and their interest in the proposed methodology.
>
> **Clarification on Conceptual Framing**:
> We agree that induction heads are not the sole mechanism behind ICL, as discussed in Section 2. Our aim is not to argue that induction heads explain all instances of ICL. Rather, we are curious about investigating whether they are necessary during training for other ICL behaviors (i.e., abstractive mechanisms) to emerge. Prior work (e.g., Olsson et al., 2022) has suggested such dependence. We added Figure A24, which shows that copying-task accuracy rises roughly 1,000 steps earlier than other tasks and coincides with the sharp loss drop in Figure A8. This timing motivates our question of whether the loss decrease and the onset of inductive copying serve as prerequisites for the emergence of abstractive ICL. Our training protocol therefore examines how learning progresses when inductive copying is suppressed and induction heads remain weaker, as outlined in the abstract. We have revised the conclusion to reflect this more precise formulation, and we thank the reviewer for their feedback
>
> **Discussion on Weakened Induction**:
> Thank you for this comment. As noted in the Abstract and Section 5, we have emphasized that induction heads are weakened. We would be glad to revise any sections where additional clarification could improve the manuscript.
>
> **Relation Between HAPAX and Memorization**:
> We agree that investigating memorization under HAPAX would be valuable! In our work, our focus is on the relationship between inductive copying and in-context learning, whereas memorization concerns whether the model stores information in-weights rather than how it learns abstractive ICL behaviors during training. Therefore, we feel that memorization is a whole new direction that would warrant a full paper in itself. Our hope is that HAPAX can directly inspire such work. Reducing copying may indirectly affect memorization, and we agree it is relevant to the broader ICL landscape, but our contribution centers on how suppressing a known attention mechanism shapes the development of abstractive ICL. Memorization in our paper appears mainly through references to prior work using loss masking to reduce private-information memorization. Although we also use loss masking, our strategy targets copying behavior rather than memorization.
>  Regarding single-exposure learning, it would require storing associations like “Token A → Token B” after one instance, which is in-weights memorization rather than in-context copying. Our inductive-copying evaluation uses randomly generated sequences, so such associations would on average be unhelpful.
>
> **Link Between Copying & Induction Mechanisms**:
> Thank you for this comment. We motivated our masking scheme as an intervention designed to reduce inductive copying (Section 3, L 99–103). In the background section, we highlighted that prior work identifies induction circuits as the dominant mechanism by which transformer models implement n-gram copying, which motivates our approach. Furthermore, Section 5 presents post-training analysis showing that once inductive copying is suppressed, the model exhibits fewer prefix-matching attention heads. We would be glad to further clarify any of these points if helpful.
>
>
> **Questions**:
> 1. Our HAPAX masking is applied only within the local window, as noted on Section 3 (L 104-106). Since our analysis does not focus on memorization, we cannot make strong claims, but prior work has shown that different loss-masking schemes can reduce memorization. Even if single-exposure learning occurs, we would expect it to be reduced relative to the vanilla model.
>
>
> 2. We interpret the reduction of prefix-matching heads as evidence that our training intervention successfully reduced inductive copying and also weakened the corresponding induction circuits.
>
>
> 3. What we refer to as abstractive tasks aligns with fuzzy induction. For instance, in translation, the target sequence does not appear verbatim in the prompt, so the model must infer the right output from the relational pattern. As shown in Figure 4 and Table 1, our model continues to perform well on these tasks.
>
>
> 4. We thank the reviewer for their sharp observation. We have considered this idea as well, but removing repetitions directly from the input would distort the structure of many sentences. For example, in the sequence [“I”, “told”, “her”, “that”, “I”, “told”, “you”], deleting the induction-predictable positions would produce [“I”, “told”, “her”, “that”, “I”, “you”]. This forces the model to learn an unnatural continuation (“I", "you”). Masking avoids this issue by keeping the input intact while simply preventing gradients at induction-predictable positions.
>
> 5. When we say the model operates *without copying*, we are referring to its functional behavior. It performs poorly on copying tasks, as shown in Figure 3.

---

### Meta-Review · Area_Chair_f6G2 · 2026-01-14

**Summary:**

The paper investigates whether induction heads are essential for ICL capabilities. The authors propose a framework that suppresses induction heads (by removing the contribution form the loss function). The performance on ICL tasks was comparable in the empirical evaluation. Given that this is primarily an experimental paper and that many concerns were raised by reviewers (and not all of them could be adequately clarified during the rebuttal period) the paper may need further work.

**Reviewer Concerns:**

The reviewers were unsure that the empirical evaluation was as thorough as it needs to be.

**Reviewer Scores:**

It is unlikely that more reviewers would have significantly increased their scores. With the exception of one review, the initial scores were very low, so even with a modest increase in review scores, the paper would not meet the bar.

---

### Decision · Program_Chairs · 2026-01-26

Reject